# Ret kinase-mediated mechanical induction of colon stem cells by tumor growth pressure stimulates cancer progression in vivo

Thanh Huong Nguyen Ho-Bouldoires [1,8], Kévin Sollier [1,8], Laura Zamfirov [1,2,8], Florence Broders-Bondon [1,8], Démosthène Mitrossilis [1,3], Sebastian Bermeo [1], Coralie L. Guerin [4], Anna Chipont [4], Gabriel Champenois [5], Renaud Leclère [5], Nicolas André[5], Laurent Ranno [6], Aude Michel [7], Christine Ménager [7], Didier Meseure [5], Charlie Demené [2], Mickael Tanter [2], Maria Elena Fernández-Sánchez [1,9✉] & Emmanuel Farge [1,9✉]

How mechanical stress actively impacts the physiology and pathophysiology of cells and tissues is little investigated in vivo. The colon is constantly submitted to multi-frequency spontaneous pulsatile mechanical waves, which highest frequency functions, of 2 s period, remain poorly understood. Here we find in vivo that high frequency pulsatile mechanical stresses maintain the physiological level of mice colon stem cells (SC) through the mechanosensitive Ret kinase. When permanently stimulated by a magnetic mimicking-tumor growth analogue pressure, we find that SC levels pathologically increase and undergo mechanically induced hyperproliferation and tumorigenic transformation. To mimic the high frequency pulsatile mechanical waves, we used a generator of pulsed magnetic force sti-mulation in colonic tissues pre-magnetized with ultra-magnetic liposomes. We observed the pulsatile stresses using last generation ultra-wave dynamical high-resolution imaging. Finally, we find that the specific pharmacological inhibition of Ret mechanical activation induces the regression of spontaneous formation of SC, of CSC markers, and of spontaneous sporadic tumorigenesis in Apc mutated mice colons. Consistently, in human colon cancer tissues, Ret activation in epithelial cells increases with tumor grade, and partially decreases in leaking invasive carcinoma. High frequency pulsatile physiological mechanical stresses thus con-stitute a new niche that Ret-dependently fuels mice colon physiological SC level. This process is pathologically over-activated in the presence of permanent pressure due to the growth of tumors initiated by pre-existing genetic alteration, leading to mechanotransductive self-enhanced tumor progression in vivo, and repressed by pharmacological inhibition of Ret.

[1] Institut Curie, Université PSL, Sorbonne Université, CNRS UMR 168, Laboratoire de Physico-Chimie Curie, Mechanics and Genetics of Embryonic and Tumoral Development team, INSERM, F-75005 Paris, France. [2] Physics for Medicine Paris, ESPCI ParisTech, PSL Research University, Inserm U1273, F-75005 Paris, France. [3] Biomedical Research Foundation of the Academy of Athens, 4 Soranou Ephessiou St., 115 27 Athens, Greece. [4] Cytometry Platform, Institut Curie, Paris, France. [5] Platform of Investigative Pathology, Institut Curie, 75248 Paris, France. [6] NEEL Institut, CNRS, Grenoble Alpes University, F-38042 Grenoble, France. [7] Sorbonne Université, Laboratoire PHENIX Physico-chimie des Electrolytes et Nanosystèmes Interfaciaux, CNRS UMR 8234, F-75005 Paris, France. [8] These authors contributed equally: Thanh Huong Nguyen Ho-Bouldoires, Kévin Sollier, Laura Zamfirov, Florence Broders-Bondon. [9] These authors jointly supervised this work: Maria Elena, Fernández-Sánchez, Emmanuel Farge. ✉email: maria-elena.fernandez-sanchez@curie.fr; emmanuel.farge@curie.fr

Mechanical stresses due to cell deformation or substrate stiffness change have been identified to actively change cell phenotypes, such as cell shape, apoptosis, stiffness, or motility[1–4]. Mechanical cues strongly influence cell fate and physiological functions[5,6], but can also induce pathological dysfunction such as amplifying hyper-proliferation and invasiveness during tumorigenesis[7,8]. These mechano-biological processes rely on the mechanical regulation of protein activities, often localized at junctions, through mechanically induced conformation changes allowing biochemical reactions that activate classical pathways[9–11].

Specifically, the role of cell deformation and substrate stiffness in cell fate and differentiation has been highlighted thus far in embryonic and stem cells, as well as in differentiated adult cells[5,6,12–14]. These properties were subsequently found as involved in many physiological embryonic processes in response to internal pressure and stress-associated morphogenetic movements[15–17]. In diseases, physical parameters characterized by abnormal mechanical stress, such as high fibrotic substrate stiffness and hyper-proliferative pressure, lead to tumorous biophysical and biochemical states that enhance invasiveness and hyper-proliferation[18–20], or to mechanically induced cell extrusion defects[21,22].

However, very few of these studies were performed in vivo[23]. The use of physical force-generating methodologies to quantitatively mimic the physiological and pathological forces involved, and unambiguously test the mechanical cue implication in the addressed processes, is even rarer[8,24]. In addition, none of these studies have, to our knowledge, linked mechanical cues involved in homeostatic process regulation to their deregulation leading to pathological features in vivo.

The colon is from that perspective of potentially high relevance. Indeed, colon tissue is constantly subjected to pulsating physiological mechanical signals involved in intestinal transit. In addition, pulsatile waves are required for the process of organogenesis which allows the folded-growth of human colon-on-chip[25]. Physiological pulsatile strains are composed of distinct movement frequencies. In rodents, the highest one, myogenic in nature and characterized by a period of 2 s, has poorly understood functions[26]. Pulsatile waves are periodic in contrast to tumor growth pressure which is permanent. We thus wonder whether pulsatile mechanical stresses could mechanotransductively stimulate the induction of SC, which maintain homeostatic renewal levels[27] through proliferation, morphogenesis, and biochemical patterning of healthy colonic adult epithelial tissue. We additionally wonder if, in the presence of tumors, initiated by genetic alterations, compressions exerted by associated hyper-proliferative cells lead to an amplification of the physiological pulsed mechanical cues, thereby pathologically amplifying SC number in the healthy tissues compressed by the tumor, and creating a permanent positive feedback loop proliferating signal further fueling tumor growth.

We focused on the highest frequency stress pulsation because it is, temporally speaking, the most intense known pulsatile strain of colon motility. It is therefore, mechanotransductively speaking, potentially the most efficient one. We found mice colon high frequency pulsatile mechanical stresses required for the maintenance of the physiological SC number, in vivo.

In mice, associated stresses lead to several hundreds of microns pulsed movements of ~1kPa amplitude at the 2 s periods (refs. [26,28] and Supplementary Note 1a), a value comparable to the 1kPa rodent permanent tumor growth pressure[8,29]. Indeed, we furthermore find that the effect of high-frequency pulsatile stresses on the stimulation of physiological SC rate is pathologically over-amplified by additional permanent anomalous tumor growth pressure, leading to a doubling of SC and proliferative cell (PC) number, as well as to the generation of CSC markers

expression in Apc mutated mice (a mutation found in 85% of human somatic colon tumors[30,31]), and to tumorigenic hyper-proliferation mechanical stimulation. Thus, high-frequency pulsatile stress cues act as a new niche for SC, through the Ret/β-cat mechanosensitive pathway. While the overactivation of the Ret/β-cat dependent SC production by additional pressure due to tumors initiated by genetic alterations, fuels tumor growth and progression. All of these processes are repressed by Ret inhibitor chemotherapeutic treatment, in vivo. We finally find Ret anomalously activated in human colon tumors, as well as in all 9 other human solid tumors tested here.

## Results

**High frequency colonic pulsatile mechanical stresses maintain physiological levels of Lgr5+ stem cells number in colon crypts, in vivo.** Lgr5 is the biomarker of colonic SC and a key factor in the colon crypt renewal by controlling the turnover of epithelial cells. Its expression is regulated by the β-cat signaling pathway[27]. Interestingly, the mechanically induced activation of the Ret kinase promotes the β-cat pathway[8] in response to a 1kPa permanent tumor growth pressure that is in the same order of magnitude of the endogenous pulsatile pulsed mechanical strains amplitude undergone by the mouse distal colon tissue. This raises the intriguing possibility that endogenous pulsatile mechanical strains participate in the maintenance of Lgr5+ cells.

To address the role of pulsatile mechanical stresses in the stimulation of Lgr5+ SC number, we first applied to Lgr5-GFP mice the cannabinoid receptor agonist WIN55,212-2 (WIN) inhibitor of colon pulsatile movements[32] for 5 days. Five days is the characteristic time of complete renewal of colonic tissue[33,34], at which any putative physiological role of pulsatile movements in renewal stress should be efficient. We observed a significant decrease of the number of Lgr5+ SC number on the order of two in peristalsis-deficient mice (Fig. 1a, b), suggesting that endogenous stresses promote Lgr5+ SC maintenance.

To investigate the involvement of peristaltic-induced mechanical stresses, we chose to focus on the high-frequency movements of a period of 2 s because of its higher temporal intensity and potential higher mechanotransductive effect. We then monitored endogenous movements in vivo by using ultrafast ultrasonic imaging in anesthetized mice. We observed, in vivo, colonic pulsatile movements of 5 s period, longer than the previously reported 2 s period[28], possibly due to anesthesia effects. The movements are characterized by an amplitude of 200 μm that is reminiscent with the several hundreds of microns amplitude observed ex-vivo (Fig. 1c-left, d-left and Supplementary Fig. 1a–c with associated Supplementary Movie 1 in which propagates the periodic waves of alternatively positive (red) and negative (blue) radial movements along the gut longitudinal axis (which center is in black), the 1 s global pulsations being respiratory beating pulsations of the anesthetized mice)[26,28]. After WIN treatment, we found the inhibition of these pulsatile movements amplitude with a reduction of a factor of 2, indicating that motility motor neurons are involved in the amplitude of high-frequency pulsations enhancement[26] (Fig. 1c-right, d-right and Supplementary Fig. 1d with associated Supplementary Movie 2). And as a result, WIN-treated mice with reduced pulsatile movements were characterized by a significant reduction of Lgr5+ SC of a factor of nearly 2 after 5 days (Fig. 1a, b and Supplementary Note 1b).

To rescue the reduced pulsatile stresses due to WIN treatment, we first magnetically loaded the colon mesenchymal conjunctive tissues with ultramagnetic liposomes (UML) after intravenous injection by using a strong small magnet localized on the skin in front of the colon for 30 min[8]. UML is labeled with a Rhodamine fluorochrome engrafted on PE phospholipids and detectable by

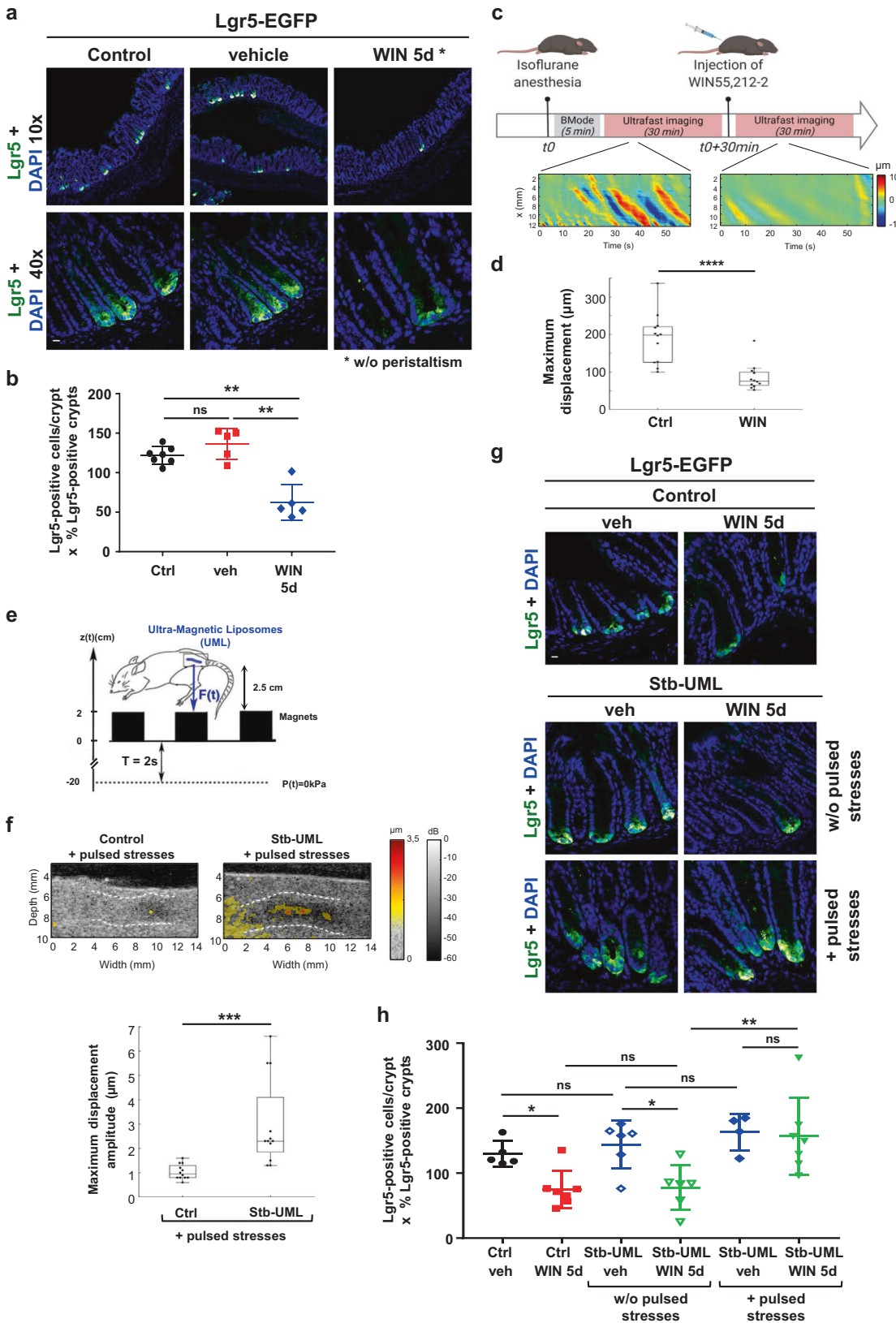

fluorescence[8]. We previously showed that UML are stable in the

mesenchymal conjunctive tissue of the colon in the presence of
the magnet for at least 1 month[8]. Here we could see the loaded
UML in the colon conjunctive tissue 30 min after injection in the
presence of the magnet and for at least 3, 5 months after magnet
removal (Supplementary Fig. 2a, b and Supplementary Note 2a).

Its presence was found in Vimentin positive mesenchymal cells
with a similar dynamics (Supplementary Fig. 2a, c). Furthermore,
cytometry analysis shows that free-circulating UMLs were washed
out from the blood plasma and from a small quantity of cells (6%)
after 3 h (Supplementary Fig. 2d, e and Supplementary Note 2b).
We thus subsequently applied a pulsed magnetic field gradient

**Fig. 1 High frequency pulsatile mechanical stresses maintain Lgr5+ stem cells physiological rate in colon crypts, in vivo. a** Visualization of Lgr5+ SC in the colon of Lgr5-EGFP mice after WIN treatment for 5 days. The number of SC Lgr5-EGFP positive cells per crypt was calculated with the images taken with a ×40 objective and the percentage of Lgr5-EGFP positive crypts in the colon with images obtained with a ×10 objective. Scale bar is 10 μm. **b** Quantification of **a**. Mean number of Lgr5-EGFP positive cells per crypt multiplied by the percentage of positive crypts in the colon. Mann–Whitney test; **$p < 0.01$, ns not significant. **c** Visualization of endogenous colonic pulsatile waves using ultrasound. Experimental set-up and timeline of the experiment. Heat maps represent the space- and time-dependent axial displacement along the colonic wall in WT mice colon before and after injection of WIN. **d** Quantification of the pulsatile activity in WT mice colon before and after injection of WIN. ($n = 4$ mice, three acquisitions were analyzed for each mouse and for each condition, before and after drug injection). The WIN acquisitions were chosen in a time window between 10 and 25 min after drug injection. Mann–Whitney test; ****$p < 0.0001$. **e** Pulsated magnetic field gradient set-up drawing. The force F(t) induced by the 2 cm magnets of 1Tesla magnetization oscillate between the force F generated at $z = 2.5$ cm and $F = 0$ at $z = -20$ cm below the colon, mimicking endogenous pulsatile deformations (see text). The mouse drawing was imported from reference[8]. **f** Up, ultrasound imaging of mouse colon deformation induced by magnetic stimulation. Representative B-Mode acoustic image of a non-injected colon sample with sur-imposed color-coded 2D map of the maximum displacement amplitude compared to magnetically loaded colon explant after magnetic stimulation. White lines represent colon walls. Down, quantification of the maximum displacement amplitude in WT colon samples injected or non-injected with magnetic liposomes after magnetic stimulation. Mean of the maximum displacement amplitude in magnetized colon explants = $3 \pm 1.8$ μm ($n = 4$ mice with three measurements per mouse). Mean of the maximum displacement amplitude in control colon explants = $1 \pm 0.3$ μm ($n = 4$ mice with three measurements per mouse). Mann–Whitney test, ***$p < 0.001$. Note that the p-value between control and Stb-UML is still significant even if we do not take into account the three higher measurements. **g** Lgr5+ SC in the colon after application of a pulsated magnetic compression mimicking high-frequency colon pulsatile movements for 5 days in the Lgr5-EGFP mouse model. Control: mice injected with UML only without magnet implantation; Stb-UML: mice injected with UML and subjected to 30 min of magnet implantation to stabilize the UML; veh: injected with the vehicle of WIN; WIN: injected with WIN; Lgr5+ cells have been detected with an anti-GFP antibody. $N = 3$ experiments. Scale bar is 10 μm. **h** Quantification of **g**. Mean number of Lgr5-EGFP + SC per crypt multiplied by the percentage of positive crypts per mouse in the colon. Mann–Whitney test; *$p < 0.05$; **$p < 0.01$; ns: not significant. Error bars: standard deviation, except for d and f in which it represents the minimum to maximum data values (excluding one outlier for d-WIN - the outlier being taken into consideration in the p-value evaluation) and for which the box represents the difference between the 75th and 25th percentile.

with the physiological period of 2s[28], for 5 days (Supplementary Note 1). This was performed thanks to a two-dimensional checkerboard made-up of 2 cm size permanent magnetic cubes of 1 Tesla, separated by 2 cm (see Methods). This ensured a relatively constant force amplitude, generated by magnetic field square gradient on the order of 0.07 $T^2$/m at the 2.5 cm distance of the colon of the mice evolving in the cage plan (Fig. 1e, Supplementary Fig. 3, and Supplementary Note 3a, b). This gradient was designed to be equivalent to the 0.07$T^2$/m gradient characteristic of a magnet already measured to generate 1kPa of pressure in the colon (Supplementary Note 3a)[8]. Pulsations of magnetically induced 1kPa pressure were thus ensured by the oscillation of a magnetic checkerboard driven by a motor, between 2.5 cm and 22 cm every 2 s for 5 days (Fig. 1e and Supplementary Movie 3). This oscillatory magnetic field gradient induced periodic pressure was detected through the rescue of a 2 s periodic pulsed movements in endogenous pulses defective mice colons, by using 2D tissue speckle tracking performed by an ultrasonic probe, (Fig. 1f, Supplementary Movie 4, Supplementary Movie 5, and Supplementary Note 3c). Movements detectable on the overall colon organ are of 2 s period. Their amplitude is consistently smaller of two order of magnitudes than endogenous pulsatile ones, because the 1kPa stress is here magnetically produced directly in the conjunctive epithelial tissue that includes epithelial crypts only. Indeed, the latter is ten times thinner than the visceral smooth muscle from which the 1kPa stress produces the endogenous peristatic movements imposed to the conjonctive and crypt tissues[35] (see Supplementary Note 3d). Thus, the magnetic treatment does not rescue the physiological amplitude and propagative nature of the overall colonic structure pulsatile movements, but rescues in the conjonctive and epithelial cells of interest the physiological pulsatile 1 kPa high-frequency stress pressure of 2 s generated by associated physiological movements.

As a response to magnetically induced 2 s pulsed 1kPa stresses in the conjonctive and epithelial crypts, we found physiological levels of Lgr5+ SC rescue in distal mice colon crypts from WIN-treated mice defective conditions (Fig. 1g, h).

These results show that mechanical stresses generated by high-frequency pulsatile movements are mechanical physiological cues

that promote Lgr5+ cells physiological production in mice colon crypts.

**The mechanical stimulation of Ret phosphorylation drives the high-frequency pulsatile-induced maintenance of homeostatic levels of Lgr5+ stem cells.** To test whether Ret mechanosensitivity underlies the mechanical stimulation of physiological levels of Lgr5+ SC in response to endogenous colonic pulsatile mechanical strains, we first investigated Ret phosphorylation state in WIN-treated pulsatile-defective and magnetically-rescued conditions.

In the colon of wild-type (WT) mice, the number of epithelial cells with physiological activation of Ret by its phosphorylation on Y1062 is between 1 to 3 cells per crypt, in nearly 10% of the crypts (Fig. 2a, b, white arrows). WIN-treated mice with reduced pulsatile movements significantly showed a 1.5-fold reduction of Ret phosphorylation after 48 h (Fig. 2a, b and Supplementary Fig. 4a, b).

We then tested if the pulsed mechanical rescue of Lgr5+ SC cells levels from WIN-treated pulsatile-defective conditions levels observed in Fig.1g, h is dependent on mechanical stimulated Ret phosphorylation, by specifically inhibiting Ret phosphorylation mechanical activation with Vandetanib (Vande) (Supplementary Note 4 and 5 with Supplementary Figs. 5 and 6 respectively). Indeed, such rescue was inhibited in mice treated with Vande Ret inhibitor (Fig. 2c, d and Supplementary Fig. 7a, b).

These results show that the high frequency pulsatile mechanical stresses are required for maintaining the physiological levels of Lgr5+ SC number in the colon crypt, through the mechanical activation of Ret (Supplementary Note 6a, b).

**Permanent tumorous mechanical stresses over-stimulate the multiplication of Lgr5+ stem cells and Notch1+ proliferative cells in vivo.** Since we found that the periodic pulsed mechanical stress mimicking high frequency physiological pulsatile movements maintains the homeostatic rate of Lgr5+ SC, we wondered whether additional permanent 1kPa pathological mechanical stresses induced by tumor growth pressure could over-activate the SC

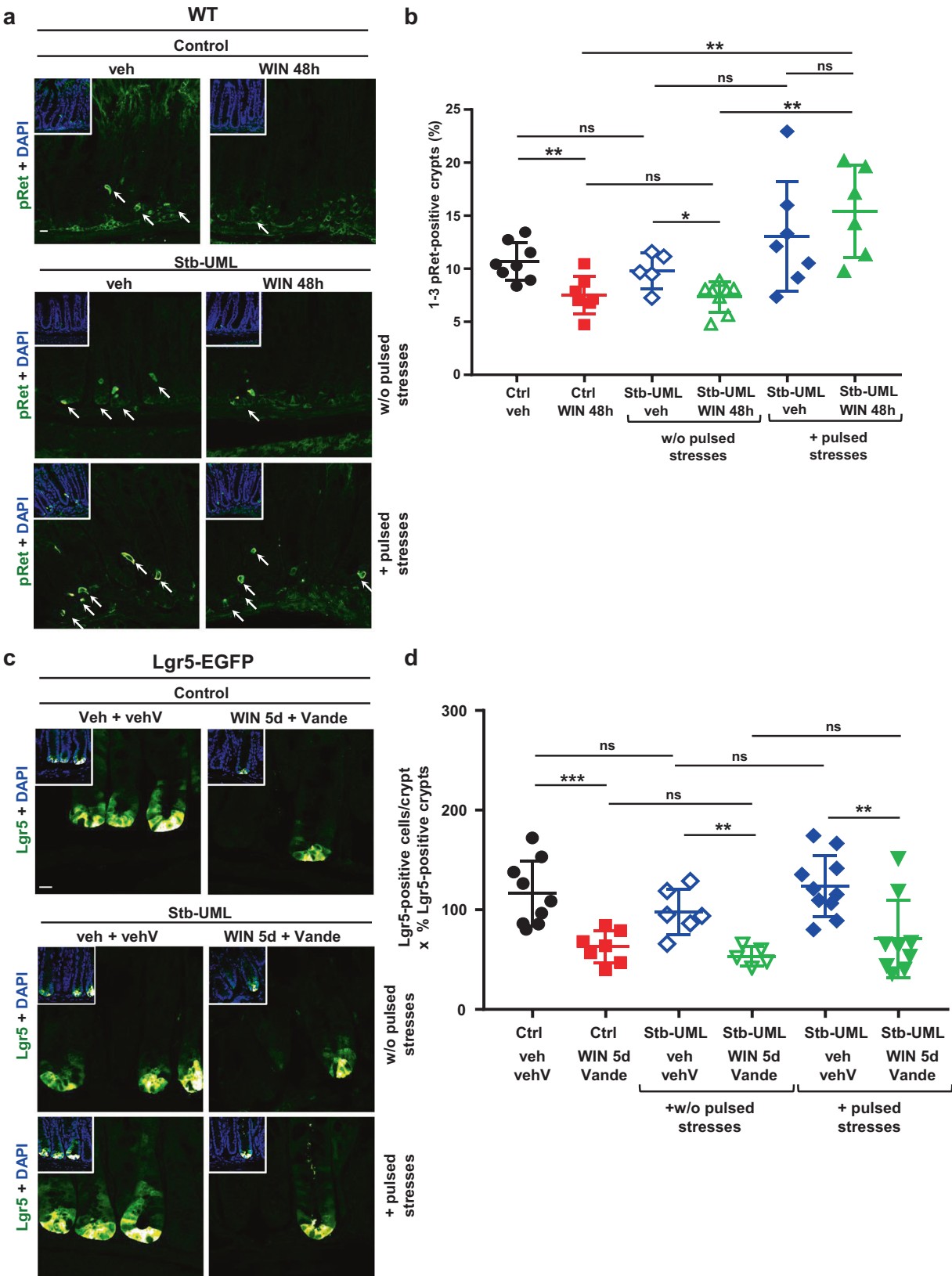

multiplication promoting the underlying cellular mechanism of tumorigenic hyper-proliferation mechanical induction. We thus magnetically applied the 1 kPa pressure with a small strong magnet localized on the skin in front of the colon after UML intravenous injection, that have been previously quantitatively measured[8] and shown to simulate hyper-proliferative Anomalous Crypt Foci

(ACF) formation that initiates tumorigenesis, after 1 month[8]. Here we found UML present and stable in colonic conjunctive tissues from 1 week[8] and up to 3,5 months in the presence of the magnet (Supplementary Fig. 8a, b). Similarly to pulsatile experiments (Supplementary Note 2), the presence of the stabilizing magnet was required for the presence of injected UML into the conjunctive

**Fig. 2 The mechanotransductive stimulation of pRet drives high frequency pulsatile colonic stresses maintenance of homeostatic Lgr5+ SC number, in vivo. a** Levels of Y1062 Ret kinase phosphorylation after colon magnetization with UML in WT mice, veh: injected with the vehicle of WIN, WIN: implemented with WIN, + Stb-UML: previous conditions with UML stabilized in the colon in the presence (+ pulsed stresses) or in the absence (w/o pulsed stresses) of the pulsated magnetic field designed to produce deformations of pulsatile movements amplitude and frequency (see Fig. 1f), for 48 h. White arrows show crypts with a pRet signal of 1–3 pRet positive cells/crypt, $n = 3$ experiments. Scale bar is 10 μm. **b** Quantification of **a**. **c** Lgr5+ SC in the colon after application of a pulsated magnetic compression mimicking high frequency pulsatile colonic stresses for 5 days in the Lgr5-EGFP mouse model, with and without Vande treatment. Control: mice injected with UML only without magnet implantation; Stb-UML: mice injected with UML and subjected to 30 min of magnet implantation to stabilize the UML; veh: injected with the vehicle of WIN; WIN: injected with WIN; vehV: implemented with the vehicle of Vande; Vande: treated with Vande. Lgr5+ cells have been detected with an anti-GFP antibody. $N = 3$ experiments. Scale bar is 10 μm. **d** Quantification of **c**. Mean number of Lgr5-EGFP+ SC per crypt multiplied by the percentage of positive crypts per mouse in the colon. Mann–Whitney test; *$p < 0.05$ **$p < 0.01$; ***$p < 0.001$; ns not significant. Error bars: standard deviation.

tissue, as observed by the absence of UML after injection without magnet (UML 1m-2m), comparable to control without UML injection and magnet (Ctrl condition) (Supplementary Fig. 8a, b). Its presence was found in Vimentin positive mesenchymal cells with a similar dynamics (Supplementary Fig. 8a, c). Consistent with the fact that UML should be in contact to the endothelial cells of the conjunctive tissue before extravasation, we here additionally checked and found the presence of UML in CD31 positive endothelial cells as well, with a similar dynamics (Supplementary Fig. 8a, d). We observed an 1.6-fold increase in Lgr5+ SC number per crypt at 1 month of mechanical stimulation of the WT mice colon (Fig. 3a, c, Supplementary Note 7, and Supplementary Note 8). A similar increase in Lgr5+ SC number was also observed after 1 month of tumor mechanical stress in the Apc heterozygous strain (Fig. 3b, d). The basal rate of Lgr5+ SC in *Apc*-mutated mice was higher than in the WT mice, which could explain the significantly higher number of Lgr5+ SC cells in the *Apc* heterozygous mice colon compared with WT mice colon after mechanical induction, of a factor of about 1.7.

We additionally analyzed Notch1 as a marker of colon crypt proliferative cells (PC). Indeed, mice possess four Notch paralogues, Notch1,2,3,4 with Notch1 and 2 only being expressed in mice gut track epithelia with redundant proliferative function, and a large prevalence of Notch1 on Notch2[36]. In addition to be expressed into mice colon, Notch1 is overexpressed in mice colon adenocarcinoma[37]. In addition to cell fate decision, crypt SC maintenance and differentiation, Notch1 is thus a major regulator of cell proliferation, and hyperproliferation in cancer[38,39]. As Notch 1 antibody was not available, we monitored the expression of membrane GFP induced by the Notch1 promotor, via Cre-Lox induction[36] (see Methods). The number of PC expressing Notch1 increased after 1 month of mechanical magnetic pressure, with an increased factor of 1.7 in WT mice and 1.5 in the Apc genetic background mice, compared with control mice non-subjected to mechanical magnetic pressure (Supplementary Fig. 9a–d).

These results show that tumoral permanent 1kPa pathological mechanical stresses significantly induces the increase in the SCs number, with final levels of SC amplified by Apc heterozygous mutation, and leads to a hyperproliferation state after 1 month.

**Permanent tumorous mechanical stresses stimulate the expression of cancer stem cells markers in Apc heterozygous mice colon, in vivo.** Cancer stem cells (CSC) population is characterized by extensive self-renewal and contributes to metastasis and chemotherapy resistance[40]. Colon CSC are multipotent neoplastic cells involved in tumoral progression, tumor growth, and recurrence since they resist to classical chemotherapy[41,42]. The conditions of their emergence, maintenance, and proliferation are therefore of major importance in cancer research. Here we thus tested if, like SC, CSC production is mechanotransductively activated by permanent pressure characteristic of tumor growth in vivo, in the predisposed Apc genetic background.

CSC markers were selected based on their expression in human colorectal tumor patients. Among colon CSC markers, Lgr5 is expressed in normal colonic tissue SC as well as in tumorous colonic tissue CSC. Its overexpression increases with cell transformation and is most elevated in colon carcinoma[43]. We found a significant increase in the number of Lgr5+ cells after 1 month of magnetically induced tumor growth pressure stress exerted on Apc 3 month-old mice colon (Fig. 3b, d).

Other CSC markers were tested: CD133/Prominin, Sox2, Aldh1/2, CD44v6, CD24, ALCAM/CD166. All of them are known to be expressed in normal cells and overexpressed in malignant cells[44–47].

For CD133, we observed positive crypts in 9-month-old Apc background mice with spontaneous colorectal tumors, compared with control 4-month-old Apc mice without any tumor visible by colonoscopy (Fig. 3e, f, Supplementary Fig. 9e, f, Supplementary Fig. 10a–c, and Supplementary Note 9). Interestingly, we observed more intense CD133 stained crypts in Apc;Lgr5-EGFP mice with gastric tumors at 5 months compared with weaker signals in control 4 and 9 month-old control mice without tumors (Fig. 3e, f).

Following 1 month of magnetically induced tumor growth pressure initiated on 3-month-old mice, we found a significant increase in the number of the CD133+ crypts per mouse compared with control mice (4 and 9-month-old), and comparable with its level in 5-month-old mice with gastric tumors showing that magnetically induced tumor growth pressure promotes the expression of the CD133 cancer cell marker in Apc colon crypts.

Additional CSC markers Sox2, CD44v6, and Aldh1/2 expression were analysed after magnetically induced permanent tumor growth pressure in vivo in Apc;Lgr5-EGFP mice. For all of these, we observed a more intense staining in control mice of 9-months-old v/s 4-months-old, and in UML Magnet mice with 1 month of magnetic pressure applied on 3-months-old v/s control mice of 4 and 9-months-old (Supplementary Fig. 10a, b).

Therefore, pathological permanent magnetically-mimicked pressure characteristic of tumor growth promotes the expression of cancer stem cells markers to pathological tumorous levels in the colon of mice with the Apc heterozygous genetic background.

**Mechanically induced increase in stem cells and cancer stem cell markers rate is dependent on the mechanotransductive activation of Ret in Apc mice colon.** As already mentioned, the SC and CSC Lgr5 marker is a direct target of β-cat[48], which is downstream of Ret[49]. We thus used the Ret inhibitor Vande to evaluate the role of Ret in the mechanically induced increase in Lgr5+ cell number by magnetically-mimicked tumor growth pressure, in the Apc heterozygous genetic background.

Strikingly, we observed a full inhibition of the mechanical induction of Lgr5+ cell number production (Fig. 4a, c), as well as a consistent full decrease in the number of mechanically induced

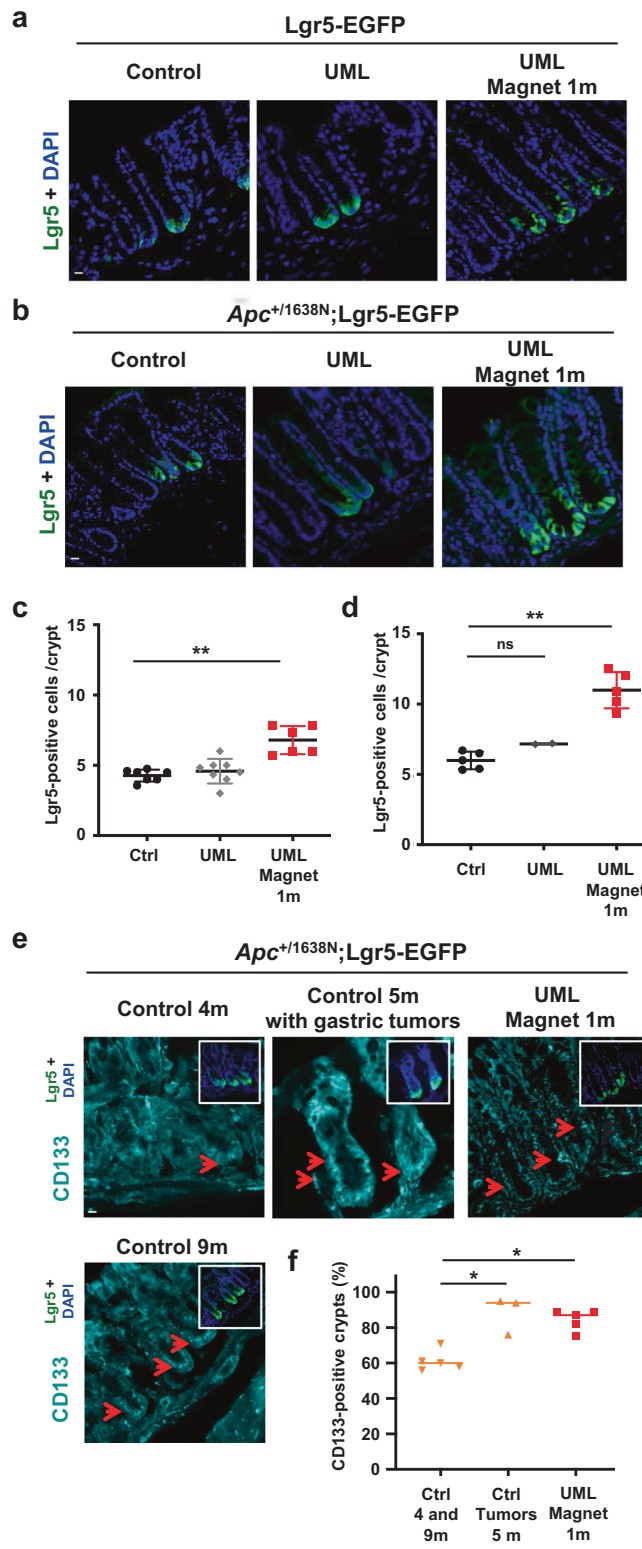

**Fig. 3 Permanent tumorous mechanical stresses over-stimulate Lgr5+ SC number and CD133 CSC production, in vivo. a** Lgr5+ SC in Lgr5-EGFP mice after application of a permanent magnetic compression mimicking tumor growth pressure for 1 month. Control: mice without UML injection ($n = 7$ mice); UML: mice injected with UML without magnet implantation ($n = 8$ mice); UML Magnet: mice injected with UML plus permanent magnet implantation for 1 month ($n = 6$ mice). Scale bar is 10 μm. **b** Lgr5+ SC in Apc;Lgr5-EGFP mice after application of a permanent magnetic compression mimicking tumor growth pressure for 1 month. Control: mice without UML injection ($n = 5$ mice); UML: mice injected with UML without magnet implantation ($n = 2$ mice); UML Magnet: mice injected with UML plus permanent magnet implantation for 1 month ($n = 5$ mice). Scale bar is 10 μm. **c** Quantification of **a**. Mean number of Lgr5-EGFP + SC per crypt and per mouse. Mann–Whitney test; **$p < 0.01$. **d** Quantitative analysis of **b**. Mean number of Lgr5-EGFP + SC per crypt and per mouse. Mann–Whitney test: **$p < 0.01$; ns not significant. **e** CD133 + CSC after application of a permanent magnetic compression mimicking tumor growth pressure for 1 month in Apc;Lgr5-EGFP mice. Controls: 4 and 9 month-old mice ($n = 5$ mice); Control 5 month-old mice having gastric tumors: small intestine and colon ($n = 3$ mice); UML Magnet 1m: mice injected with UML plus permanent magnet implantation for 1 month ($n = 5$ mice). Red arrows show CD133 + crypts. Small white frames show similar images with Lgr5-EGFP and nuclei staining. Scale bar is 10 μm. **f** Quantitative analysis of **e**. Percentage of CD133 + crypts per mouse. Mann–Whitney test; *$p < 0.05$. Error bars: standard deviation.

following Vande treatment (Fig. 4a, d and Supplementary Fig. 11a, c). Strikingly, the spontaneous overexpression of CD133 in 16 month-old Apc mice was also inhibited by Vande treatment (Fig. 4e, f).

These results show that the mechanical induction of SC and of CSC by pathological tumor growth pressure is a process triggered by Ret mechanical induction, with CSC induction being Ret dependent as well during spontaneous progression towards tumorigenesis, in the Apc heterozygous context.

**Paneth niche cells of the intestine are mechanically induced in Apc heterozygous mice colon in a pRet mechanosensitive dependent process.** The biochemical microenvironment also plays a key-role in determining SC function and fate, constituting a cellular niche indispensable for the maintenance and balance keeping between self-renewal and differentiation of SC[51]. We next examined whether a permanent mechanical stress could, in addition to direct mechanically induced production of more SC in the crypts, also have a stimulating effect on the cellular niche of colonic SC.

We did not observe any change in the number of Gli1+ niche cells in the mesenchyme[52], after 1 or 3.5 months of mechanical compression (Supplementary Fig. 13a, b). We observed no increase, rather a decrease in RegIV+ cells, previously described as deep crypt secretory niche cells (DCS) in the colon[53,54] (Supplementary Fig. 13a, c). Paneth cells also sustain an essential SC niche in mice intestinal crypts[55]. More rarely found in colon crypts[54–56], they are abnormally observed in human colon cancer tissues[57]. We thus checked whether Paneth cells could be present as SC niche cells mechanotransductively induced by tumor growth pressure into the colon. We observed an increase of 3.7% in the number of lysozyme positive Paneth cells in the colon after 1 month of magnetically induced permanent tumorigenic mechanical stress (Fig. 5a, c). Interestingly, mechanical induction of Paneth cells production was even stronger after 3.5 months of mechanical stress, where an 8% increase was detected, a level consistent with the 13% observed in human colorectal cancers[57] (Supplementary Note 10).

Notch1+ PC number increase (Supplementary Fig. 11a, b), following treatment with Vande. Coherently, the mechanical induction of the Ki67 proliferative marker expression[50], which was comparable to spontaneous tumorous Ki67 levels, was inhibited by Vande treatment as well (Supplementary Fig. 12a–d).

In addition, we importantly found a full decrease in the expression of CD133 CSC marker induced by pathological permanent magnetically-mimicked tumor growth pressure,

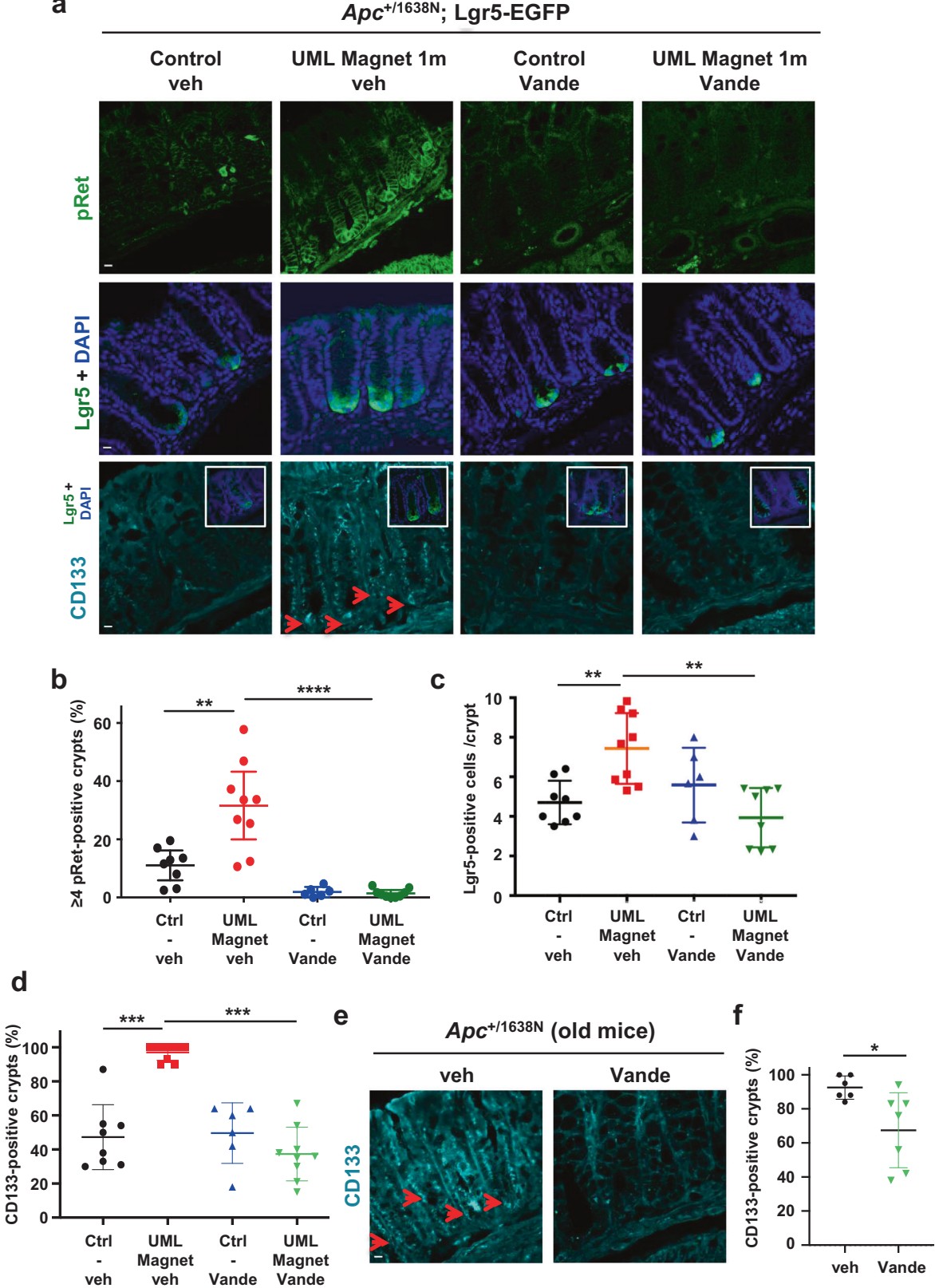

**a** *Apc*[+/1638N]; Lgr5-EGFP

This interestingly showed the mechanical induction of the intestinal niche cells of Lgr5+ SC by magnetically-mimicked tumor growth pressure, in the colon.

We then wondered whether the induction of Paneth cells observed in response to permanent tumorigenic mechanical stress is dependent on pRet mechanical stimulation. We found that the treatment with pRet inhibitor Vande completely abolishes mechanical induction of Paneth cells in response to magnetically produced 1 month-tumor growth pressure (Fig. 5b, d).

Therefore our data showed that Paneth niche cells usually only found in the intestine are mechanically induced in the colon by

**Fig. 4 The mechanical stimulation of SC and CSC multiplication is pRet dependent in Apc heterozygous mice colon, in vivo. a** Levels of pRet+ crypts (up), Lgr5+ SC (middle), and CD133 + CSC (down) in Apc;Lgr5-EGFP mice after application of a permanent magnetic compression mimicking tumor growth pressure for 1 month, with and without Vande. Control Veh: mice without UML injection treated for 1 month with vehicle of Vande ($n = 8$ mice); UML + Magnet 1 m Veh: mice injected with UML plus permanent magnet implantation and treated with vehicle for 1 month ($n = 9$ mice); Control Vande: mice without UML injection treated for 1 month with Vande ($n = 6$ mice); UML + Magnet 1 m Vande: mice injected with UML plus permanent magnet implantation and treated with Vande for 1 month ($n = 8$ mice). $N = 2$ experiments. Red arrows show CD133 + stained cells. Small white frames show similar images with Lgr5-EGFP and nuclei stainings. Scale bar is 10 μm. **b** Quantitative analysis of the pRet signal. Percentage of crypts with ≥4 pRet positive cells per mouse. Mann–Whitney test; $**p < 0.01$ and $****p < 0.0001$. **c** Quantitative analysis of Lgr5-EGFP signal. Mean number of Lgr5+ SC per crypt and per mouse. Mann–Whitney test two-tailed; $**p < 0.01$. **d** Quantitative analysis of CD133 signal. Percentage of CD133 + crypts per mouse. Mann–Whitney test; $***p < 0.001$. **e** CD133 + cancer cell marker in Apc and Apc;N1Cre-ERT2 old mice (16 month-old). CD133 antibody staining in colon crypts of mice treated with vehicle ($n = 6$ mice) and Vande ($n = 7$ mice). Red arrows show CD133 + stained cells. Scale bar is 10 μm. (**f**), Quantification analysis of **e**. Percentage of CD133 + crypts per mouse. Mann–Whitney test. $*p<0.05$. Error bars: standard deviation.

## The pharmacological inhibition of the Ret/β-cat pathway inhibits mechanical induction of Anomalous Crypt Foci formation stimulated by tumor growth pressure, in vivo. 

Pathological tumor growth pressure triggers Ret activation through the phosphorylation of its Y1062 site. Such activation was in vivo correlated with the activation of the β-cat pathway via the Y654-βcat phosphorylation target of Ret, cytosolic accumulation of β-cat and expression of β-cat tumorigenic target genes including *myc, axin-2* and *zeb-1*, hyperproliferative ACF tumorigenesis initiators in the Apc mice. But no *direct* causal relation was experimentally tested in vivo between these mechanically induced events and Ret phosphorylation mechanical activation (Supplementary Note 11)[8]. Here we find that the mechanical activation of Ret by tumor growth pressure is upstream of the β-cat target gene Lgr5 in SC and CSC in vivo, thanks to the use of the Vande specific inhibitor of Ret mechanical induction of phosphorylation in mice colon (Fig. 4a, c). We thus used the Ret inhibitor Vande to test for the causal link between the mechanical activation of Ret phosphorylation induced by tumor growth pressure, the mechanical activation of the β-cat pathway and the tumorigenic colonic ACF production in vivo.

First, we checked that the pathological overactivation of pRet by magnetically induced tumor growth pressure, characterized per crypts with more than the physiological 1–3 cells displaying pRet (≥4 pRet+ cells, Supplementary Fig. 5), was inhibited by chemotherapeutic treatment with Vande from the beginning of the imposed pressure of 2 h to 1 month, on Apc mice (Fig. 4a, b and Supplementary Fig. 14a, b). We then found an inhibition of both the mechanically induced pY654-β-cat phosphorylation and β-cat cytosolic accumulation after 1 month-treatment with Vande in Apc;Lgr5-EGFP mice (Supplementary Fig. 14c–e).

This showed Ret mechanical activation by magnetically-mimicked tumor growth pressure as inductive of the β-cat pathway activation upstream of the SC and CSC mechanical induction (Fig. 4).

Downstream of the Ret-dependent SC and CSC mechanical induction should thus be a Ret-dependent tumorigenic hyperproliferation mechanical induction. Under normal conditions, 3–5 month-old Apc mice display a little number below 7 of ACF, *i.e* crypts with aberrant cell colon proliferation characterized by unorganized irregular blue coloration shape compared with regular crypt area (see orange arrows, Fig. 6a, c). When Apc mice were subjected to colon permanent tumor growth mechanical pressure, the pre-tumorous ACF number per mouse initiated a significant increase after 2 weeks comparing with control Apc mice[8], with twice to 3 times more ACF than without mechanical stimulation after 1 month of permanent mechanical pressure

(Fig. 6a, b). This process was inhibited by daily treatment with Vande in Apc heterozygous Lgr5-EGFP mice (Fig. 6a, b), as well as in Apc heterozygous and Apc;Notch1-GFP genetic background mice (Supplementary Figs. 15 and 16).

The specific role of Ret in the β-cat dependent increase in Lgr5+ cells mechanical induction leading to ACF formation was further confirmed by using Danusertib (Danu), another inhibitor of Ret activation[58]. Interestingly, Danu treatment did repress mechanical stimulation of pRet, with a slightly lower efficiency than Vande (Supplementary Fig. 17a, b and Fig. 4a, b). Consistently, the mechanical stimulation of pβcat, cytosolic βcat and Lgr5 expression downstream of pRet were repressed by Danu treatment, with a sliglthty lower efficiency than with Vande as well (Supplementary Fig. 17a, c, d, e, Supplementary Fig. 14, and Fig. 4a, c). However, inhibition of pRet by Danu was sufficient to efficiently inhibit mechanical induction of ACFs similarly to Vande (Supplementary Fig. 18a, b). The inhibition of Lgr5 expression induction in response to ex-vivo direct 1kPa mechanical stimulation[59] in the presence of Danu, like in the presence of Vande (Supplementary Fig. 18c, d) confirms the specific role of Ret in the β-cat dependent mechanical indution of the Lgr5 positive SC at the orgin of ACF formation in response to magnetically-mimicked tumor growth pressure in vivo.

Second, to test whether Ret mechanical induction by tumor growth pressure indeed plays a role in endogenous tumor progression, we treated 16 month-old *Apc* heterozygous mice exhibiting sporadic spontaneous ACFs with Vande. We found both a decrease in Ret anomalous activation and in ACF number after one month of treatment (Fig. 6c, d and Supplementary Fig. 19a, b). Even though we didn't observe a reduction of more advanced colon polyps by Vande treatment (Supplementary Fig. 19c, d), we however noticed a pronounced tendency towards a reduction in small intestine sporadic spontaneous tumors with Vande taking (Fig. 6e, f), possibly reflecting the efficient absorption of Vande at the intestine reducing its efficiency into the colon.

Overall, these data converge to the fact that tumor growth pressure plays a major role in tumoral progression, through the Ret/β-cat mechanical activation inducing an increased rate of SC and CSC, and that tumorigenesis can be thwarted by pharmacological treatment with Ret inhibitors of tumorigenic mechanical induction.

## Y1062-Ret phosphorylation is activated in human colon and other human solid tumors. 

Since Ret is expressed in human colon epithelial cells and our data have shown its mechanical activation by tumor growth pressure in mice, we wondered whether Ret could be also activated in human colon tumors. Indeed, we found the presence of Y1062-Ret phosphorylation from the earliest to the most advanced stage of tumorigenesis with a grade mean value of about 2 to 3 compared to 0 in the WT colon (3 being the maximal score) (Fig. 7a). Furthermore, we

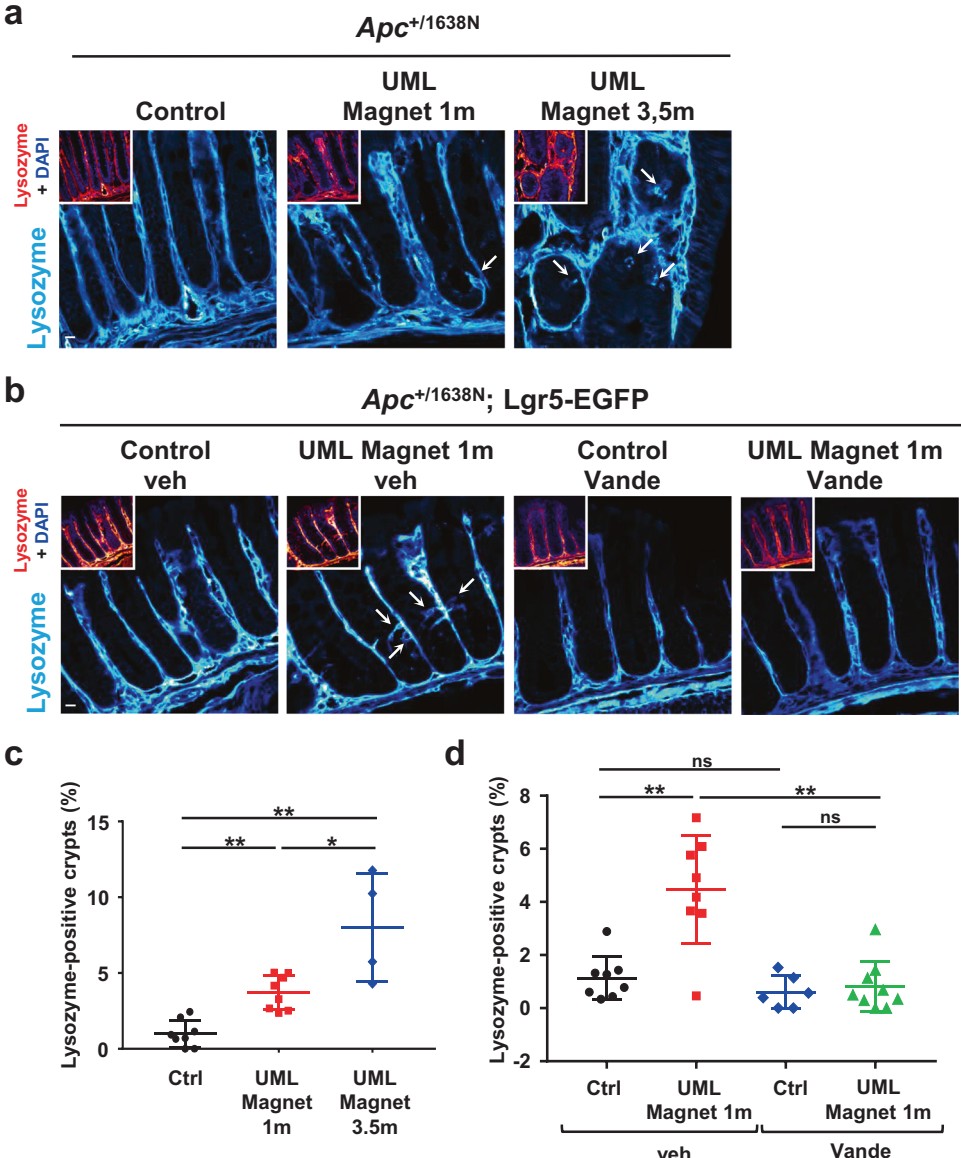

**Fig. 5 Paneth niche cells of the intestine are mechanically induced in Apc heterozygous mice colon in a Ret mechanosensitive dependent process.**
**a** Lysozyme staining of the colon tissue after application of a permanent magnetic compression mimicking tumor growth pressure for 1 month ($n = 8$ mice) and 3.5 months ($n = 4$ mice), compared to control ($n = 8$ mice) in Apc mice. Small boxes display lysozyme positive signal in red to increase contrast with DAPI in blue. Scale bar is 10 µm. **b** Lysozyme staining of the colon tissue after application of a permanent magnetic compression mimicking tumor growth pressure for 1 month, treated with the pRet inhibitor Vande in Apc;Lgr5-EGFP mice. Control Veh: mice treated with the vehicle of Vande ($n = 8$ mice); UML Magnet 1 m Veh: mice injected with UML plus permanent magnet implantation for 1 month and vehicle treated ($n = 8$ mice); Control Vande: mice treated with Vande for 1 month ($n = 6$ mice); UML Magnet 1 m Vande: mice injected with UML plus permanent magnet implantation for 1 month and Vande treated ($n = 9$ mice). Small boxes display lysozyme positive signal in red to increase contrast with DAPI in blue. **c** Quantification of **a**. Mean number of lysozyme positive crypts per mice. Mann–Whitney test; *$p < 0.05$; **$p < 0.01$. **d** Quantification of **b**. Mean number of lysozyme positive crypts per mice. Mann–Whitney test; **$p < 0.01$; ns not significant. Error bars: standard deviation.

expected to observe an activation of Ret dependent on the tumor growth pressure in the human colon tumoral tissue. As this tumor growth pressure should increase with tumor stage and partially decrease after the leaking of the primary tumor at the invasive stage, we actually observed the increased Ret activation with the tumor progression grade from 0 in the WT tissue to 3 in the early invasive intra-mucosal adenocarcinomas, and a 1.5-score lower activation in colon adenocarcinomas infiltrating the submucosa, the muscularis and/or the subserosa (Fig. 7b).

Tumor growth pressure is, in its physical nature, a parameter that should also be present in all solid growing tumors. We would

thus expect a generic activation of Ret phosphorylation in most primary solid tumors types. Indeed, Ret was found activated in primary tumors at invasive stage in the pancreas, ovary, lung, head and neck, uveal, breast, endometrium, with a positive tendency for the uterus cervix, with scores varying between 0.2 to 2.2 depending on the organ (Fig. 7c).

Alike for mice colon tumor progression, these results indicate human tumor growth pressure as inductive of Ret activation in the human colon. They additionally suggest such behavior as generically involved in most of the 9 other organs solid tumors tested.

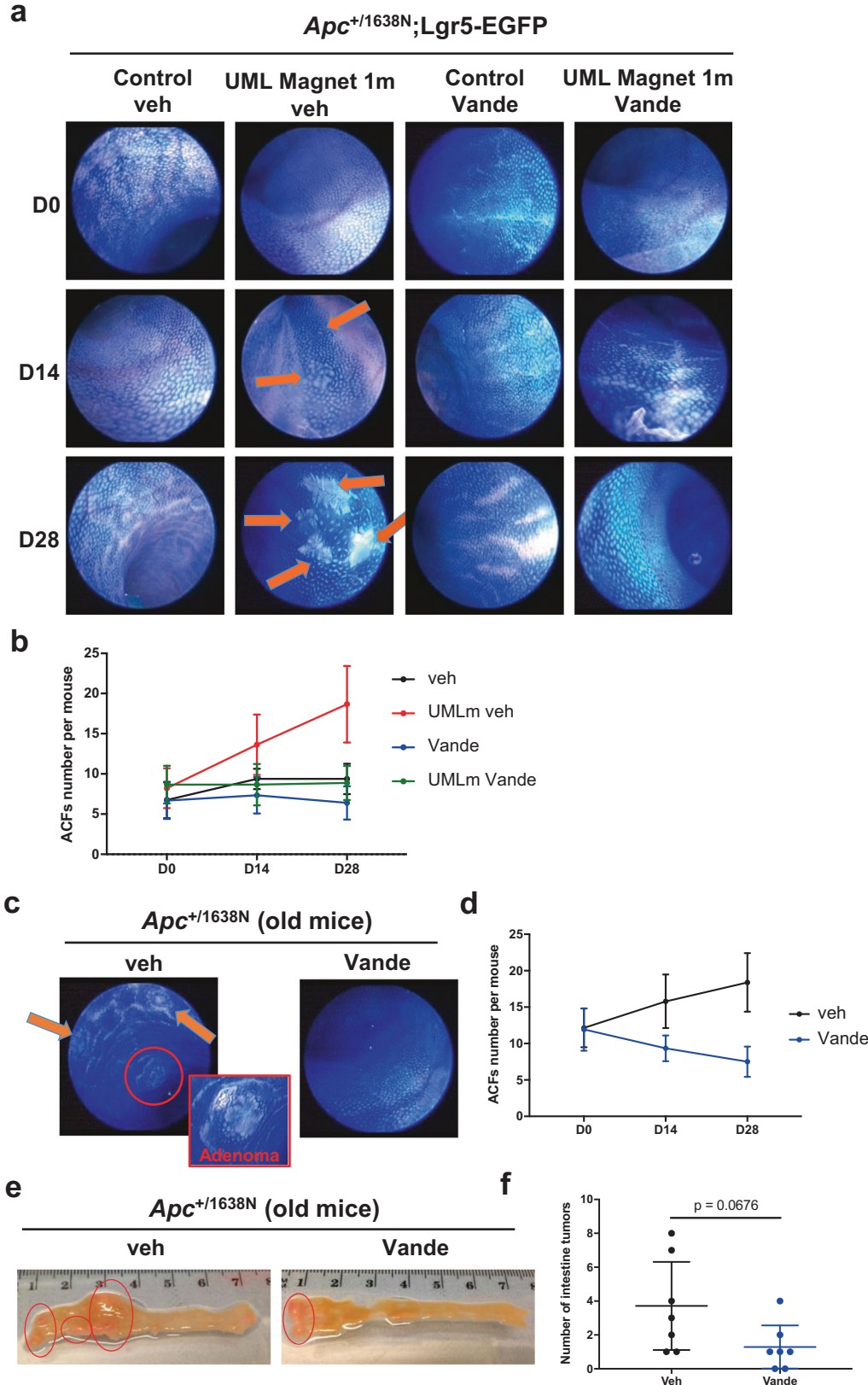

## Discussion

The presence of a tumor, initiated by pre-existing genetic alterations, generates abnormal permanent mechanical stresses that consist of a mechanotransductive activator of pathological pathways and tumorigenic processes. These can act inside the tumor[7,60], as well as outside the tumor in the healthy tissue compressed by tumor growth[8]. This is the case for mice colon tumorigenesis[8], through the anomalous mechanical reactivation of the embryonic β-cat dependent pathway involved in mouse fetal villi developmental gut organogenesis[61], a pathway also known to be mechanosensitively activated in early endomeso-derm specification by embryonic gastrulation morphogenetic

**Fig. 6 The pharmacological inhibition of the Ret/β-cat pathway with Vande inhibits mechanical induction of, and spontaneous tumorigenesis initiation in vivo. a** Live imaging of ACFs (orange narrow) at different times of the experiment subjected to the application of a permanent magnetic compression mimicking tumor growth pressure for 1 month, with or without Vande treatment in Apc;Lgr5-EGFP mice. **b** ACFs number counting at different times until 28 days of the application of permanent magnetic compression mimicking tumor growth pressure application on the colon of Apc;Lgr5-EGFP mice with or without Vande treatment, $n = 6–9$ mice/condition. $N = 2$ experiments. Statistical significance determined using the Holm-Sidak method, with alpha = 0.05. Adjusted p (D14, UMLm veh vs. veh) <0.05, adjusted p (D28, UMLm veh vs. veh) <0.01, adjusted p (D14, UMLm vande vs. UMLm veh) <0.05, adjusted p (D28, UMLm vande vs. UMLm veh) <0.01. **c** Live imaging of ACF (orange arrows) at 28th day of vehicle or Vande treatment of Apc old mice. **d** ACF number counting at different times during the 1 month of vehicle or Vande treatment, $n = 13–14$ mice/condition. $N = 3$ experiments. Statistical significance determined using the Holm-Sidak method, with alpha = 0.05. Adjusted p (D28) <0.01. **e** Intestine upper section pictures after 1 month of vehicle or Vande treatment. Intestine tumors are surrounded by a red circle. **f** Quantification of **e**. $N = 7$ mice/condition. Statistical significance determined using Mann–Whitney test. Error bars: standard deviation, except for **d**, **f** in which it is standard error to the mean.

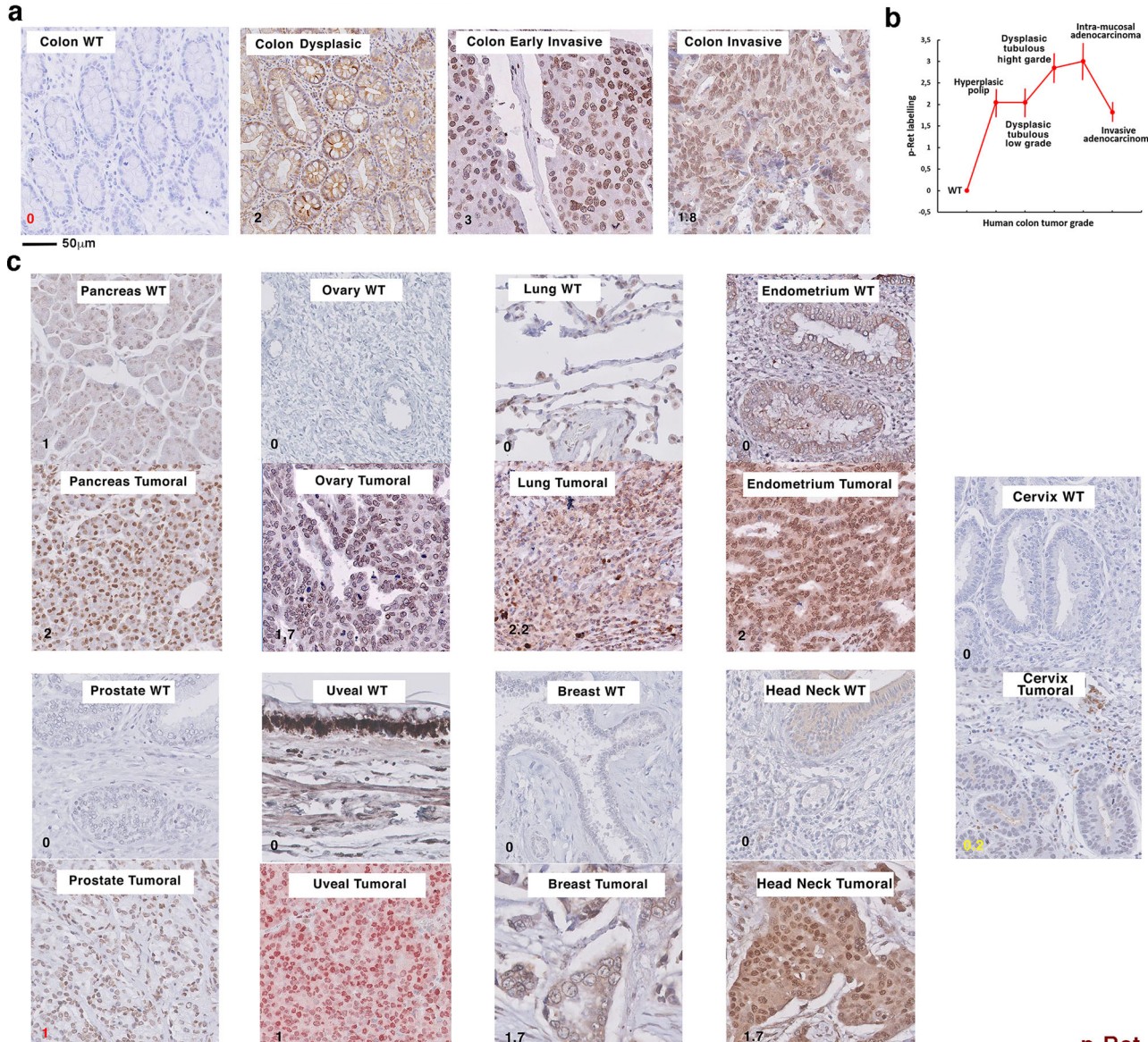

**Fig. 7 pY1062-Ret is activated in human colon, and other human solid tumors. a** Y1062 phosphorylation of Ret (pRet) in the colon of WT (N=10), dysplastic tubulous adenoma ($N = 10$), early invasive intra-mucosal adenocarcinoma ($N = 10$) and invasive (sub-mucosa and beyond) ($N = 10$) adenocarcinoma of variable grade ($N = 10$). **b** Level of activation of pRet as a function of the tumorous grade ($N = 10$ by stage). Error bars: standard deviation. **c** pRet in adenocarcinoma invasive primary tumors of pancreas ($N = 71$), ovary ($N = 101$), lung ($N = 31$), endometrium ($N = 31$), prostate ($N = 31$), uveal ($N = 61$), head and neck ($N = 61$), breast ($N = 61$), and cervix malignant ($N = 51$) tumors. All WT controls are $N = 10$ (see adapted statistical analysis procedure in methods). Numbers correspond to the phosphorylation level order of magnitude in its predominant subcellular location. Scores in black: $p < 0.01$, in yellow: $p = 0.25$ tendency, in red: data heterogeneity prevents statistical analysis.

movements in other species like Drosophila and zebrafish[24,62]. The question thus comes why evolution did not eliminate such mechanosensitive-associated fragility in the maintenance of organ homeostasis regulation, to prevent animals from any mechanical sensitivity potentially leading to hyper-proliferation and tumorigenic process in its adult colonic tissue.

In the Drosophila embryo, for instance, the expression of the APC/GSK3β degradation complex of cytoplasmic β-cat begins after the earliest stages of embryonic development when gastrulation morphogenetic movements mechanotransductively stimulate endoderm specification[63]. This thus acts de facto as a protective system, buffering any pathological mechanotransductive expression of embryonic endodermal gene expression potentially due to mechanically induced accidental abnormal cytoplasmic enrichment of β-cat, after gastrulation and at later stages in adult gut epithelia.

On the other hand, adult tissues physiological homeostasis is ensured by a steady state dynamical equilibrium between a β-cat-dependent reminiscent development-like regeneration processes, due to SC activities and apoptotic processes[21,64]. Indeed, the inhibiting role of APC/GSK3β on the β-cat pathway activation is now known to be specifically removed in SC, though the action of Wnt which is expressed by niche cells and which inhibits GSK-3β[65,66], making of SC local regenerative embryonic-like developmental cells in the adult tissue. In addition, mechanical inductive cues acting through the mechanotransductive activation of the β-cat pathway can therefore act on the downstream pathway of β-cat independently of Wnt expression[24].

This thus asks the plausible question of the conservation of the role of mechanical stresses in the activation of the β-cat pathway from gut early developmental stages to the maintenance of the SC developmental-like activities in adult mice gut colonic tissues, to which adds Wnt expression by niches cells. Indeed, the gut is the object of natural physiological mechanical stresses, which consist in pulsatile mechanical stresses. Here we thus specifically raised the question of a role of the pulsatile mechanical stresses in the maintenance of colonic SC activities.

By pharmacologically reducing the amplitude of the high-frequency pulsatile stresses by a factor of 2 with WIN, and magnetically rescuing it via pulsed mechanical stresses of 2 s period in vivo, we found the respective decrease by a factor of 2, and full rescue of the physiological Lgr5+ SC number in WT mice colon. This process dependents on the mechanical activation of the Ret kinase (Supplementary Note 12a). Gut pulsatile movements have many functions, including gut transit motility driven by the slaw 1 min period pulsations, and microbiota maintenance[67]. However, unless they are ubiquitously found in many distinct species, the physiological function of high-frequency colonic pulsatile pulsations has remained until now poorly investigated[26]. Here we find high frequency pulsatile mechanical stress as a cue required for the physiological maintenance of SC number, through the mechanical activation of Ret (Supplementary Note 12b).

As mechanical induction of hyperproliferation and tumorigenesis by tumor growth pressure is correlated with the activation of the mechanosensitive Ret kinase in vivo[8], here we thus tested whether the later process could be due to an abnormal over-activation of the Ret mechanosensitive pathway, and of the number of SC downstream, by pathological permanent tumor growth pressure mechanical stresses, added to the endogenous pulsed physiological pulsatile stresses.

Indeed, we found the magnetically-mimicked permanent tumor growth pressure as Ret-mechanical-activation-dependent inducer of a pathological 1.5-fold increase in the number of SC and PC into mice crypts, leading to a pathological level of SC per crypt in the Apc heterozygous context representative of 85% of human colon tumors, with levels nearly twice higher than in uncompressed WT. In experiments with pulsatile movements, strains and signaling levels were enough to detect the highly sensitive Ret mechanosensitive signal, but not the downstream β-cat known as much less sensitive[68,69]. In contrast, here we consistently detected the Ret mechanical activation-dependent tumorigenic increase of β-cat cytosolic concentration in response to the more intense tumor growth pressure added to endogenous pulsatile stresses in the Apc heterozygous context. We additionally found that the induction of CSC markers expression is Ret-mechanical-activation-dependent in predisposed Apc heterozygous mice background crypts.

This demonstrates that the homeostatic Ret mechanical activation is dependent on physiological mechanical strains of pulsatile movements. And that it is amplified by additional abnormal pressure characteristic of the growth of tumors, that pathologically trigger an increase in SC number, thereby leading to hyperproliferation in the WT, and to CSC markers expression and tumorigenesis stimulation in the Apc heterozygous context (Supplementary Note 13–14).

Because we here find that pulsatile mechanical stresses are required for maintaining physiological levels of SC number, like Wnt signaling by niche cells[56] and because mechanical induction is more efficient in Apc deficiency contexts[8,24], we suggest that both the inhibition of Apc-dependent degradation of cytoplasmic β-cat, ensured by Wnt expression by niche cells, and the upstream of β-cat signaling Ret mechanical activation, ensured by pulsatile mechanical stresses, are required for SC specification and number physiological levels in the WT mice. Increasing the intensity of mechanical stresses due to pathological tumor growth pressure, which adds to pulsatile movements, mechanotransductively increases the physiological Ret/β-cat pathway and downstream the number of SC leading hyperproliferative anomalies in the WT. Furthermore, by decreasing the level of Apc of 50% in the Apc heterozygous mice, all crypt cells, including SC, are even more sensitive to mechanical stresses by degrading less efficiently mechanically induced and Wnt induced cytosolic levels of β-cat. This over-activates a developmental-like process of mechanical activation of β-cat dependent gene expression generating new SC up to tumorigenic number levels. The same mechanism was observed with CSC.

Indeed, in addition to the requirement of an Apc mutation for promoting the mechanical induction of tumorigenesis, a permanent 1kPa pressure applied for months represents an anomalous mechanical environment of higher intensity compared to the physiological 1 kPa pulsed mechanical stresses, integrated over time. As the physiological number of SC was found to be mechanosensitively maintained by endogenous pulsed mechanical stresses, the additional permanent tumoral stress should consequently lead directly to a significant increase in SC number, which we observed. This increase in turn should lead to cell hyperproliferation initiating tumorigenesis in compressed tissues, due to the increase of crypt cells renewal dynamics as a direct consequence of the SC number increase that generate proliferative cell (PC) number increase, both of which we also observed. We propose that this process amplifies the progression of spontaneous sporadic tumors initiated by genetic alteration, by adding mechanical strains leading to more production of SC in healthy tissues compressed by neighboring tumors, in the endogenous process of tumor progression. A process that should stimulate spontaneous sporadic tumor progression of endogenous tumors on the Apc mutated background old mice, which should be repressed by the inhibition of the Ret mechanosensitive pathway. Which we additionally observed.

The permanent 1kPa pressure applied for months also represents a relatively high anomalous mechanical context as well

compared to benign transient strains that can accidentally be applied to colonic tissues (i.e., transient constipation).

In conclusion, we found that high frequency pulsatile mechanical stresses – or visceral muscle cells leading to pulsatile stresses - are one of the niches required for mice colonic SC homeostatic production, together with the Gli1-expressing mesenchymal cells that Wnt-dependently repress APC/GSK3 dependent degradation of cytosolic β-cat[52], and make crypt cells responsive to Ret/β-catt mechanical activation and SC production. The enhancement of mechanical stress characteristic of anomalous permanent growth pressure of tumors initiated by genetic alteration adding to, and more intense than, physiological pulsatile mechanical stresses, amplifies this process and leads to hyperproliferative pathological rate of SC, which becomes tumorigenic within the Apc heterozygous context that enhances crypt cells sensitivity to SC inductive mechanical stresses. Sporadic tumors are initiated by mutations, which in Apc heterozygous mice (representative of 85% of colon tumors in humans[30,31]) is thought to be due to the full loss of APC expression[70,71] after sporadic loss of heterozygosity (in the 16-months-old mice here studied). We thus propose that, subsequently and as a second step, the stress developed by the genetically initiated tumor growth stimulates and fuels tumor progression through mechanical induction of additional SC formation in compressed neighboring healthy tissues.

Finally, and interestingly, the fact that magnetically-mimicked tumor growth pressure anomalously induces the production of the Paneth intestine niche cells into the colon, like in human colon tumors, is in line with the hypothesis that tumor growth pressure is a major effector of tumor growth progression in human colon cancer. Moreover, the Retmechanosensitive tumorigenic pathway, is anomalously activated correlatively with tumor stage in human colon tumors and generically in all primary tumor types tested at invasive stage. We suggest then the tumorigenic Ret mechanical activation as a new potential therapeutic target in tumor progression inhibition, which we here experimentally began to confirm as efficient in spontaneous colon and intestinal cancer in *Apc* heterozygous mice. Based on the preclinical improvements of systemic Ret inhibitors, their clinical trials[72] could bring high benefits to existing treatments in cancer therapies, including against CSC-dependent resistance.

## Methods

**Experimental model and subject details**. Mouse lines used include Apc$^{+/1638N}$ [73], Lgr5-EGFP-ires-creERT2 kindly provided by Philippe Sansonetti (B6.129P2-Lgr5$^{tm1(cre/ERT2)Cle}$/J Mus musculus RRID:IMSR_JAX:008875[48]), and Notch1-CreERT2/Rosa26mTmG kindly provided by Silvia Fre[36]. In the double transgenic Notch1-CreERT2/Rosa26mTmG, after tamoxifen injection, Notch1+ cells show membrane-associated green fluorescence (mG) that marks Cre-targeted cells while membrane anchored Tomato red fluorescent protein (mT) is expressed in all other cells[36]. GFP expression was here followed by immunofluorescence labelling. Apc$^{+/1638N}$;Lgr5-EGFP-ires-creERT2 and Apc$^{+/1638N}$;Notch1-CreERT2/Rosa26mTmG mice were obtained by crossing Apc$^{+/1638N}$ mice with Lgr5-EGFP-ires-creERT2 and Notch1-CreERT2/Rosa26mTmG mice, respectively. C57Bl/6J inbred mice were obtained from Charles River Laboratories. Maintenance and care of the animals was done in the Animal Facility of Institut Curie (license number D75-05-18) in an EOPS (Exempt Of Pathogen Species) controlled environment (12 h light-dark cycle, food and water *ad libitum*). Animal care and use for this study was performed in accordance with the recommendations of the European Community (2010/63/UE) for the care and use of laboratory animals. Suffering to the animals has been kept to a minimum. We comply with the established principles of replacement, reduction, and refinement. Experimental procedures were specifically approved by the ethics committee of the Institut Curie CEEA-IC #118 (Authorization reference APAFIS#10977-201708211557193 v3 given by the national authority French Ministry of Research in compliance with the international guidelines). Mice were 12–14 weeks of age at the start of experiments. Males and females were used equally. Mice with a magnet implanted on the back were kept single housed for 1 month during the in vivo magnetic deformation experiments, daily supervised, and the cage accordingly enriched. As previously described, mouse lines carrying the inducible Cre recombinase were injected intraperitoneally with tamoxifen 50 µg g-1 of animal body weight (MP Biomedicals) 24 h before sacrifice[8].

**Preparation of ultra-magnetic liposomes**. Solutions of 1,2-dipalmitoylsn-glycero-3-phosphocholine (DPPC) 25 mg/ml, 1,2-distearoyl-sn glycero-3-phosphocholine (DSPC) 25 mg/ml, 1,2-distearoyl-sn-glycero-3-phosphoethanolamine-n-[(carboxy(polyethyleneglycol)2000](ammonium salt) (DSPE-PEG2000) 25 mg/ml, and L-α-phosphatidylethanolamine-N-(lissamine rhodamine sulfonyl B) (ammonium salt) (rhodamine-PE) 1 mg/ml in chloroform were purchased from Avanti Polar Lipids, Inc. UML were prepared by the reverse phase evaporation method established in[74] and modified according to a previously described protocol[75]. In brief, a mixture of DPPC 90 µL/DSPC 10 µL/Rhod-PE 45 µL/DSPE-PEG2000 20 µL (85,2/8,8/1/5 mol%) was dissolved in 3,3 ml of diethyl ether (Sigma) and 1 ml of chloroform (Sigma). Thereafter, 1 ml of citrated magnetic nanoparticles (described in Fernandez-Sanchez et al.[8]) dispersed in the buffer was introduced before sonication at room temperature for 20 min to produce a water-in-oil emulsion. Organic solvent was evaporated with a rotavapor R-210 (Buchi) at 28 °C until the gel phase disappeared. Liposomes were filtrated through a 450-nm filter and purified from non-encapsulated magnetic nanoparticles by magnetic sorting using a strong magnet (strong magnet (NdFeB 150 × 100 × 25mm, Calamit). The operation was repeated two times every 24 h and liposomes (highly concentrated at 30% of nanoparticles in volume) were finally separated from the supernatant and recovered.

**Imaging Cytometry by ImageStream®**. UML injections and magnet placement were performed as previously described. Five minutes before blood collection, mice were anesthetized with a 80 mg/kg ketamine (IMALGENE®)/2% xylazine (ROMPUN®) mixture. Whole blood samples from aortic vein were collected in EDTA precoated-syringe with a 25-gauge needle and gently mixed in EDTA-precoated tubes.

Whole blood tubes were first centrifuged 5mn at 500 × $g$, 4 °C. Supernatants were retrieved in new tubes and centrifuged 20 mn at 5000 × $g$, 4 °C. Platelet-free plasma (PFP) were then collected for plasma fraction analysis. Pelleted blood cells from whole blood tubes were washed from red cells through a two-step lyses with BD Lysing Solution (349202, BD Bioscience) for 3mn and 7mn at RT on gentle agitation, followed by a centrifugation at 500 g, 5mn, 4 °C. White blood cells WBCs pellet were subsequently resuspended in 30 µL of FACS buffer (10% FBS, 2 mM EDTA in PBS) for WBCs analysis.

Rhodamine+ UML were quantified in PFP and WBCs by imaging flow cytometry with an Imagestream® MKII and INSPIRE® software. For both sample types, lasers 561 nm (200 mW) was used to excite Rhodamine+ UML collected in Channel 03 (Ch03) and laser 785 nm (28.07 mW) was used for Side Scatter (SSC) measurement collected in Channel 06 (Ch06). Brightfields were analyzed with dedicated channels from each camera in Channels 01 and 09 (Ch01 and Ch09). Acquisitions were performed using the 60x magnification with flow rate at low speed for high sensitivity.

Data analysis was performed with IDEAS® software version 6.2. The same analysis was applied to PFPs and WBCs samples. UML positive events were identified with a Spot Mask as events with a spot of bright intensity of Rhodamine (Spot(M03, Ch03 LIPO, Bright, 1, 2, 1)), and separated based on their intensity of SSC (MC or NMC (combination of all fluorescence = MC and the inverse of the combination of all fluorescence = NMC)). Circulating UML were identified based on their low or negative SSC ("UML + Low SSC"). WBCs associated to UML were first identified based on their high SSC ("UML + High SSC") but as this gate may still contain few events constituted of UML in coincidence with cells, debris or beads (from the cytometer) in the same field of view, a second selection was performed based on the Area of Ch01 and the Aspect Ratio Intensity of Ch01 to select "Cells" events only. This last gating strategy was as well applied to all acquired events in order to assess the total number of "Cells" (UML- and UML+) in the sample.

**Magnetic fields**. The magnetic induction (B-field in Tesla) is created by a lattice of permanent magnets. The magnets are 2 cm-cubes located in the xy plane on a square lattice. The spacing between magnets is also 2 cm, which corresponds to a 25% filling factor for the lattice. The magnets were modeled as uniformly magnetized cubes with magnetization $\mu_0 M = 1$ T, magnetized along the z-direction. The magnetic induction B, which can also be called the stray field, was calculated using an analytical approach using an electrostatic analogy. Explicit expressions (Engel-Herbert, 2005) were used to calculate the field, its norm and its gradients on the relevant meshed plane parallel to the lattice plane. The plane is located at altitude z measured from the top surface of the magnets.

**Mechanical deformations**. In vivo magnetic deformation. Mice were anaesthetized with isofluorane (delivered at 3% for induction and 1.5% for maintenance in oxygen) and a 3 mm diameter 1.4 T magnet (0.12 T measured at 2 mm) positioned subcutaneously on the back of the mice in front of the colon[8] (Calamit, Disque-NEO50 3 × 2). The fluorescent UML (associated with rhodamine-PE phospholipids) were diluted in a final volume of 100µl with 10 mM HEPES pH 7.4, 20 mM Na$_3$Citrate, 108 mM NaCl buffer solution to a final concentration of 0.2 M

and injected in the lateral caudal vein[8]. To perform the experiments mimicking pulsatile stress, the magnet was removed 30 min after implantation and served to UML stabilization only (Stb-UML). To perform the experiments mimicking permanent tumor growth pressure stress, the magnet was left for days to months in place[8].

In vivo pulsated magnetic deformation. We performed in vivo magnetic pulsed compression experiments with a custom-built system (Supplementary Fig. 3a). The mice, previously injected with UML (Stb-UML), were placed in an immobile plastic housing with food and water ($n = 5$ mice maximum by plastic cage). The transient magnetic gradient application was applied by the 0,5 Hz vertical oscillation of an engine-driven plate covered with an $8 \times 13$ array of $2 \times 2 \times 2$ cm neodymes magnets between 22 cm and 1.5 cm distance from the bottom of the cage (Supplementary Movie 3). Mice were injected with the cannabinoid receptor agonist WIN or vehicle solution when necessary. WIN 55,212-2 (WIN) was first prepared in 100% DMSO and then diluted in a vehicle solution of DMSO 10%, Tween-80 10%, and phosphate-buffered saline (PBS) 80% final volume (vehicle solutions were added in the order listed, with vigorous vortexing between steps). Drug solution was always prepared fresh immediately prior to use and administered daily via intraperitoneal (i.p.) injection in a single volume of 3 μg/g of body weight.

Ex vivo deformation. To apply an ex vivo mechanical compression colon samples were treated as previously described in[59]. Briefly, distal colon segments of ~2 cm were dissected from adult mice, rinsed with PBS and incubated at 37 °C in L15-GlutaMAX medium supplemented with 0.5% fetal calf serum, 0.08% gentamicin. Colon tissues were longitudinally opened, placed villi up in a custom-made tissue box of 3 mm thickness and a 2 cm² coverslip placed over the tissue, weighted to compress the tissue to the thickness of the box depth. The box was then submerged in the medium. Colon tissues were immediately fixed in 3% formaldehyde after a determined compression time.

**Ultrasonic analysis in vivo**. C57BL/6 mice ($n = 4$) were used in this study (12–16 weeks, 20–25 g). Mice were anesthetized with isoflurane (delivered at 3% for induction and 1.5% for maintenance in oxygen), depilated over the abdomen and placed on a 37 °C heating pad (World precision instruments). Ultrasound images were acquired using a high-frequency ultrasound probe (15-MHz central frequency, 0.11-mm pitch, 128 elements, Vermon, France) driven by an ultrafast imaging device (Vantage 256, Verasonics, USA). The ultrasonic probe was mounted on a house made motorized positioning setup enabling three degrees of translation and one degree of rotation (Physik Instrumente) (Supplementary Fig. 1a, created with BioRender.com). Ultrasound gel was applied between the ultrasound probe and the mouse abdomen to ensure proper acoustic coupling. A BMode imaging sequence was first used to localize and select a sagittal plane of the mice colon, after anal injection of a water gel transparent to ultrasound in the colon lumen in order to visualize the organ. To monitor the propagation of pulsatile waves, ultrafast ultrasound imaging was performed. Ultrasound plane-wave transmissions with $n = 6$ tilted angles (from −5° to 5° by steps of 2° tilt angle) were fired at a 3-kHz pulse repetition frequency during 60 s. Sets of $n = 6$ backscattered echoes recorded for each tilted transmission were then beamformed and coherently compounded on the fly using an in-house graphics processing units–based beamformer to produce high-quality ultrasonic images at a 500 Hz frame rate. These stacks of ultrafast images were then saved for further offline processing. Ultrafast imaging was performed during 1 h, before and after intraperitoneal injection of the inhibitor of colonic pulsatile waves WIN.

*Post-processing of in vivo acoustic analysis.* From these acquisitions, the local axial tissue displacements were estimated by using phase correlation between compounded images with a 6 ms delay: a complex pixel value of an image n is multiplied by the complex conjugate value of the same pixel in the n + 3 image later in the image stack, before taking the argument φ of the result. The tissue velocity $v$ is then estimated as $v = \lambda.\varphi/(4\pi.T)$, λ being the wavelength and $T$ the time between 3 images. Integrating this tissue velocity with respect to time yields the local displacement field. A high-pass filter and a directional filter (Deffieux et al.[76]) were applied to respectively remove global slow tissue motion and keep only the antegrade movements corresponding to the pulsatile waves. Space-time representation of these displacements was then built using a segment corresponding to the colon wall and by averaging the signal in 0.3-mm window perpendicular to that segment. An apparent slope on that space-time representation meant that a phenomenon is propagative. The maximum displacement amplitude along the temporal dimension was retrieved. Then the mean of the 20% of the highest displacement amplitude values was computed. Based on these results, for each mouse three acquisitions in a time window between 10 and 25 min after drug injection were compared to three control acquisitions of the same mouse for the quantitative analysis. Besides, the formal proof of the presence of a propagative phenomenon is provided by a power density dispersion diagram (k-ω diagram, Supplementary Fig. 1c), calculated as the squared magnitude of the Fourier transform along each dimension of the space-time displacement map (therefore given in μm².mm.s). In this diagram, power density at certain frequencies (ω) associated with a nonzero wave vector k means that this energy propagates.

**Ultrasonic analysis ex vivo**. Distal colon explants of WT mice injected ($n = 4$) or non-injected ($n = 4$) with magnetic liposomes and previously injected with

WIN55,212-2 were embedded in an agar-gelatin phantom (2% agar with 5% gelatin). Water gel transparent to ultrasound was injected in the colon lumen in order to visualize the organ. The ultrasonic probe was fixed on a frame completely independent of the magnetic stimulation device to prevent any noises from vibrations.

Ultrasound images were acquired as in acoustic analysis in vivo. Ultrasound gel was applied between the ultrasound probe and the agar-gelatin to ensure proper acoustic coupling. A B-Mode imaging sequence was first used to localize and select a sagittal plane of the colon. To monitor tissue deformations, ultrafast ultrasound imaging was performed. Several acquisitions were acquired before and during the magnetic stimulation.

*Post-processing of ex-vivo acoustic analysis.* From these acquisitions, the local axial tissue displacement amplitude was computed by using phase correlation between compounded images with a 6 ms delay: a complex pixel value of an image n is multiplied by the complex conjugate value of the same pixel in the n + 3 image later in the image stack, before taking the argument φ of the result. The tissue velocity $v$ is then estimated as $v = \lambda.\varphi/(4\pi.T)$, λ being the wavelength and $T$ the time between 3 images. Integrating this tissue velocity with respect to time yields the local displacement field. A bandpass filter (0.05–1 Hz) was applied to keep only the displacements due to the magnetic stimulation. The maximum displacement amplitude along the temporal dimension was then retrieved. Then the mean of the 10% of the highest displacement amplitude values was computed in regions of interest, i.e. within the colon or the phantom.

**Pharmacological inhibitor treatment in vivo**. The kinase inhibitors Vandetanib and Danusertib (Cliniciences) were respectively dissolved in the vehicle (1:9) DMSO/Cremophor EL® (Merck) and 5% Dextrose (Sigma). The mice, orally (50 mg/kg Vandetanib) or by IP (15 mg/kg Danusertib) and daily treated for the time of the experiment (up to 1 month), were divided into four groups ($n = 5$–14/group): veh (200 μL vehicle for Vandetanib; 100 μL vehicle for Danusertib); UML Magnet veh (mice injected with 0,2 M UML and subjected to 1kPa magnetic field, vehicle); Vande (Vandetanib, 200 μL) or Danu (Danusertib, 100 μL); UML Magnet Vande or Danu (mice injected with 0,2 M UML and subjected to 1kPa magnetic field, 200 μL Vandetanib or 100 μL Danusertib). Endoscopy and ACFs counting were performed at D0 before the beginning of the treatment and one day before the end of the experiment. For 1-month experiment, mice were also checked at D14 2 weeks after the beginning of the treatment. Mice colon were then collected, fixed and stored at −80 °C until use.

Endoscopy and ACFs counting were performed at D0 before the beginning of the treatment and one day before the end of the experiment. For 1 month experiment, mice were also checked at D14 two weeks after the beginning of the treatment. Mice colon were then collected, fixed, and stored at −80 °C until use.

**Pharmacological inhibitor treatment ex vivo**. For Ret kinase inhibition ex vivo experiments, Vandetanib (10μM), Danusertib (4μM) or an equivalent volume of the vehicle DMSO, was added to the medium with colon explants 20 min prior to the start of compression.

**Endoscopy and ACF analysis**. Anaesthetized mice were injected in the colon with warmed-up 0.5% methylene blue (Sigma) 5 min before visualization with a small-animal colonoscope. Live imaging was then performed during some minutes depending on the mouse size using a Karl-Storz endoscopic system. The camera was introduced in the rectum and a certain amount of air was injected inside the colon allowing to visualize the whole distal colon while mice were asleep on the heating pad. Image recording was done with Pinnacle Studio program. ACF counting was done during the endoscopy time and another counting after recorded Supplementary Movie visualization in a second time.

**Immunofluorescence and histology in mice**. Dissected colonic tissues were rinsed in ice-cold 1× phosphate-buffered saline buffer (PBS), fixed in 3% FA for 1 h at RT and cryoprotected in 30% sucrose in 1× PBS overnight at 4 °C. Tissues were then embedded in OCT compound (VWR), frozen, and stored at −80 °C. Immunofluorescence was performed on 6- to 8-μm thickness cryosections using standard protocols. Basic antigen retrieval incubation (R&D Systems BioTechne) was required for the staining of RegIV. Antigen unmasking solution, citrate-based (Vector laboratories) were required for the staining of Vimentin. Primary antibodies and dilutions used were as follows: Rabbit anti-Vimentin (1 :200, Sigma), rabbit anti-CD31 (1:100, Novus Biologicals), rabbit anti-pY1062 Ret (1:100, Santa Cruz Biotechnology), mouse anti-β-catenin (1:50, BD Biosciences), mouse anti-pY654 β-catenin (1:50, Santa Cruz Biotechnology), rabbit anti-Ki67 (1:200, Abcam), rabbit anti-pY1086 EGFR (1:100, Thermo Fisher Scientific), rabbit anti-pY1175 VEGFR2 19A10 (1:100, Cell Signaling), mouse anti-lysozyme (1:200, Abcam), goat anti-RegIV (1:20, Thermo Fisher Scientific), goat anti-Gli-1 (1:100, R&D Systems), chicken anti-GFP (1:2000, Abcam), rabbit anti-CD133 (1:100, Abcam), rabbit anti-CD44v6 (1:100, Sigma), mouse anti-Aldh1/2 (1:50, Santa Cruz Biotechnology), and rabbit anti-Sox2 (1:100, Ozyme). Appropriate secondary antibodies were used: anti-rabbit Alexa 488 (1:200, Molecular Probes), anti-goat Alexa 488 (1:200, Jackson ImmunoResearch), anti-chicken Alexa 488 (1:2000, Abcam), anti-rabbit

Alexa 594 (1:200, Thermo Fisher Scientific), anti-mouse Alexa 594 (1:200, Jackson ImmunoResearch), anti-goat Alexa 594 (1:200, Jackson ImmunoResearch), anti-rabbit Alexa 647 (1:200, Invitrogen), and anti-mouse Alexa 647 (1:200, Invitrogen). All antibodies were validated by publications or providers, and cited by the provider associated to the reference introduced in the up-cited table. Sections were cover-slipped with ProLong Gold Antifade (Thermo Fisher Scientific). Images were taken with a Zeiss LSM880 microscope at the Platform for Cell and Tissue Imaging or a Nikon A1R 25HD confocal microscope at the Nikon Imaging Centre located in Curie Institut. Image processing was performed using Fiji/ImageJ software.

**Immunofluorescence and histology in humans**. Routinely fixed, paraffin-embedded, 3-μm-thick tissue sections were de-paraffinized, rehydrated and then unmasked in target retrieval solution: pRet at pH 9 (30 min, 95 °C). Primary antibody detection by immunoperoxidase technique and DAB chromogenic substrate revelation on the BOND RX using the Bond Polymer Refine Detection Kit (Leica) according to the protocol recommended by the manufacturer: after blocking endogenous peroxydase activity and inhibiting non-specific staining, the slides were incubated with diluted antibodies: anti-pRet (Tyr1062) (Abcam, 1/50) overnight at 4 °C. Tissue sections were then washed with PBS and incubated with Poly-HRP IgG reagent to localize rabbit antibodies. Immunoreactive signals were detected using hydrogen peroxide substrate and 3, 3′-diaminobenzidine tetra-hydrochloride (DAB) chromogen. Finally, the sections were lightly counterstained with Mayer's Hematoxylin 3 min.

Key source table. See Supplementary Table 1.

**Statistics and reproducibility**. Quantitative results of acoustic data were analyzed using the non-parametric Mann–Whitney test, two-sided with Prism et BiostaTGV-online https://biostatgv.sentiweb.fr/?module=tests algorithms. To assess significance, we considered $p < 0.05$ statistically as the threshold for significant differences. Immunostaining and endoscopy data were analysed using the Mann–Whitney test two-sided or the Holm-Sidak method as specified in figure legends. Quantification of β-catenin signal was realized using IntDen (Integrated Density) measure function of ImageJ for each freehand cell selection. The measurements of 12 cells per base crypt were added and the mean of a minimum of 10 crypts was calculated to obtain the mean β-catenin IntDen signal per mouse.

Sample size was empirically increased until an initial tendency in the experiments could, be confirmed by a $p$-value < 0.05. Experiments were generally stopped around 10 sample by condition maximum, whatever the result. No data exclusion was performed.

Error bars are standard deviation in all experiments, expect in Fig. 1d,f in which it represents minimum to maximum data values, and for ACF measurement experiments in which it is standard error to the mean.

No blinding was performed. The many experiments realized in this study made it highly complex to manage. However, experiments were analysed in parallel by two independent investigators on the different elements of the same pathway to check for convergence of the results. No randomization was performed. Many distinct experiments were performed basically addressing the same concept with different control parameters, overall de facto excluding covariates (7 Figures, 19 supplementary Figures, each Figure including several panels): for instance, blocking mechanical strains by WIN (parameter pharmacological in nature), and rescuing it with magnetic forces (parameter physical in nature, independent of pharmacological chemistry).

Statistical analysis of pRet signaling pathway activation in human biopsies (Fig.7) was realized with Fisher paired by group of 10 samples analysis[69,77], compared to the necessarily limited 10 WT human sample available. All experiments were replicated independently (not the same day) at least twice (number of replicates N is reported in the legends, with the total sample size n being directly visible in figures by counting dots number; each dot represents a measurement in one sample - i.e. one mouse)). All other statistics were performed with the Mann–Whitney test, two-sided.

**Ethical compliance and committee**. Experimental procedures were specifically approved by the ethics committee of the Institut Curie CEEA-IC #118 (Authorization reference on APAFIS#10977-201708211557193 v3 given by the national authority French Ministry of Research in compliance with the international guidelines.

Human Biopsies: Extensive library of human colon, and other organs, cancer biopsies of the pathology service of the Institut Curie hospital, were used. These were obtained at Curie Institute and from multi-center Departments of Pathology from 1978 to 2010. For each type of tumor, ten specimens of adjacent normal tissue were used as control at RNA and protein levels. Patients treated in our institution have given their approval by signed informed consent. All patients all met the following criteria: primary tumor for which complete clinical, histologic, and biologic data were available. Quality of the data management is ensured by CDT label of the Institut Curie by the Inca, Clinic French Ethical Committee (Agreement number D-750602, France) and the ethics committee of the Institut Curie (Agreement number C75-05-18). The outcome tested was the activation of the mechanosensitve Ret phosphorylation in samples, monitored by specific phospho-antibodies y. Biopsies cross-cuts were classically generated and labeled by

the Institut Curie Pathex, with a level of labeling semi-quantitatively evaluated from 0, 1, 2 to 3. Statistical analysis were performed based on these analysis by comparison with WT samples, with a $p$-value being addressed to any element of the pathway tested in any type of tumor and grade tested.

**Reporting summary**. Further information on research design is available in the Nature Research Reporting Summary linked to this article.

## Data availability
The data sets generated and analysed during the current study are available from the corresponding authors on reasonable request.

The list of figures having associated raw data is:

Figures 1, 2, 3, 4, 5, 6, 7 and Supplementary Figs. 1–19, all panels except illustrative drawings.

Associated Source data are in Supplementary Data 1.

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

## Acknowledgements

We thank Silvia Fre for fruitful discussions. This work was funded by the FRM (grant Équipe labellisée FRM 2015 DEQ20150331702, grant Équipe labellisée FRM 2019 EQU201903007805), the Inca (PLBIO-13-172 and 2019-1-PL BIO-03-1), the ANR (16-CE14-002801), the LabEx Cell(n)Scale (grants ANR-11-LABX-0038, ANR-10-IDEX-0001-02), Q-life (ANR-17-CONV-0005). DMi is founded by the Hellenic Foundation for Research and Innovation (H.F.R.I. 2nd call for the support of Postdoctoral Researchers, project number 01281, FA 94-2/14.10.2020. Images including mice in Fig.1c and of Supplementary Fig. 1a are created with BioRender.com. We thank the members of the Animal House Facility of Institut Curie, particularly Sonia Jannet, Virginie Dangles-Marie, Mickael Garcia, Céline Daviaud, Aude Robert and Hélène Gautier, as well as Rémi Fert and Eric Nicolau from the UMR168 workshop for the setting up of the pulsed

magnetic field device. We greatly acknowledge the PICT-IBiSA@BDD and the Nikon Imaging Center NIMCE@Institut Curie-CNRS Imaging Facility of the Institut Curie.

## Author contributions

T.H.N.H.B. performed pharmacological experiments in response to tumor growth pressure & in old mice including ACF colonoscopy and pRet/pBcat IF analysis, ex-vivo direct mechanical short-time deformation and EGFR and VEGFR IF analysis; K.S. and M.E.F.S.: pulsatile experiments including pRet/Lgr5 IF labeling and analysis; LZ and MEFS: ultrasonic experiments and analysis with CD and MT; F.B.: stem cells and cancer stem cells marker experiments in response to tumor growth pressure, as well as UML localisation in IF and Ki67 proliferative, S.B.: labeling and analysis of EGFR and VEGFR in old mice experiments; D.Mi with EF built and designed the pulsed magnetic device; R.L., G.C., A.N. with D.Me labeled human samples and K.S. statistically analysed it; L.R. produced the magnetic simulations; A.M. and C.M. produced and provided the UML; M.E.F.S. produced niche cell and cytosolic β-cat labeling experiments, flux cytometry experiments with C.L.G., A.C., and K.S., as well as ex-vivo direct mechanical deformation in Ret kinase inhibitor treated experiments; and M.E.F.S. with E.F. coordinated the work and wrote the manuscript, which was implemented by all co-authors.

## Competing interests

The authors declare no competing interests
