## [Peer Review File · Communications Biology]

Reviewers' comments:

Reviewer #2 (Remarks to the Author):

The involvement of physical stresses in cancer initiation is gaining interest in the research community, but little is known about its importance or mechanisms. The authors use in situ and ex vivo analyses to study the effects of mechanical forces on colon stem cell and tumor dynamics. They conclude that high frequency pulsatile mechanical stresses maintain colon stem cell levels through Ret kinase. They propose that SC levels are pathologically over-stimulated by the physical forces, causing hyperproliferation and contributing to tumorigenesis. The work is comprehensive and potentially significant. However, there are a few issues that need to be addressed to improve the manuscript and support the conclusions.

1. In general, the text is difficult to follow and could benefit from extensive editing.
2. The experiments involving injection of the Stb-UML particles need further description. What is the fate of these particles after injection and "focusing" using the external magnet? Do they stay in blood vessels or extravasate into tissue? What are the dynamics of their clearance, and how is this affected by the magnetic forces? In the experiments where the magnet is used to localize the UMLs, and then removed, do they later disperse/disappear? And is this different in the animals where the external magnet is left in place?
3. Along these lines, there should be controls for the stb-UML particles (without magnetic forces applied) in all of the relevant main figures. It appears that many of these critical control groups are included in the extended figures, but they really need to be in the same figures with the magnet application groups.
4. In the videos, the perturbations caused by breathing appear large, and at least comparable to the peristaltic motion. Do the authors have an estimate of how the magnitudes compare? And what about normal ambulatory motion, which may also impinge on the colon?
5. The authors should show that the Ret inhibitor Vandetanib is specific, and not causing off-target effects. Another inhibitor or genetic knockout model would help.
6. The overall hypothesis is not clearly stated. Are the authors proposing that the mechanical forces imposed by peristaltic motion in normal physiology cause tumors, or that they enhance tumor growth after the tumor has already been initiated? Or are they saying that tumor growth induces forces that further enhance tumor growth? These concepts are confounded throughout the text.
7. There are two findings in the study. One shows that the forces increase stem cells, and the other shows that forces enhance tumor formation. More discussion is needed about how these are related.

Reviewer #3 (Remarks to the Author):

Ho-Bouldoires et al presented an in vivo study about the effect of mechanical stimulation, such as pulsatile stresses, on the presence and proliferation of CS and CSC cells. Moreover, they provide a mechanism for the mechanotransduction process. In specific, they point out to the Ret kinase mechanosensitivity.

My major comments are listed below:

1. While the entire investigation is quite interesting and new, the manuscript is hardly readable

due to long sentences, errors (within the sentences) and complicated description of the results. This should be improved.

2. The pulsed magnetic force stimulation of colonic tissues that have been magnetized with ultrasound-magnetic liposomes seems to be intriguingly. However, I do not understand how the intravenously injected liposomes enter the targeted cells, which seems to be mesenchymal cells in the colon. How do they get there in a targeted manner? Has it been checked where these liposomes are? Are they all only in these cells and why not in blood cells or endothelial cells, since they get in contact with them much earlier? Please discuss these points and provide data how you checked this.

3. Page 2 Line 5 from bottom: "This makes of high frequency..." Please rephrase. What does it mean here? (It needs to be past.)

4. Page 3 Second Para: "submitted" seems not to be the right term here. Please change.

5. Page 3 Line 11 from bottom: The sentence "Pulsation of ...motor driven... by a motor".

6. Page 3 Line 7 from bottom: There are several long sentences, such as "Movements detectable.." which needs to be rephrased.

7. Figure 1e is hardly readable (alter the colors of the text)

8. How do I know where the liposomes are in the mice when injected into the blood system? Are they marked with a fluorescent dye to follow their uptake? Which cell type harbor these liposomes? Please provide an image.

9. Page 4 Fourth Para. it should be mentioned again what WIN-defective means here. Do you mean WIN treatment? It needs to be consistent throughout the manuscript.

10. Page 4 Para 5: Please correct the sentence "Consistently, we did.... where (?)"

11. Page 4 Last Para: "By magnetic cells applying..." needs to be two sentences (hard to read).

12. Page 5 First Para: Which Notch is meant here? There are several types. Please explain why this specific one is selected then. Please provide the antibody employed (it is not in the list or provided elsewhere). It is also not given in "Extended Data Fig 7a-d".

13. Page 5 Para 3: What means "Both of them..." there are listed more than two. Please clarify.

14. Page 6: It is not explained why Paneth niche cells are investigated (no connection to the former part).

15. Page 7: "Sup Info9": I am sorry, but I cannot find it. Is it "Extended Data Fig. 9"?

16. Page 7 First Para Line 8: "We found beyond that..." Please rephrase this sentence to increase its clearness.

17. Page 8 Second Para: "Consistently, p-beta-cat..." needs to be rephrased for clearness.

18. Regarding tables 1, 2 and 3, I would show the images as supplementary data or delete this statement in the manuscript.

19. Page 8 Third Para: "We would thus expect for a...". Please cancel "for" and check all other sentences for errors, such as typos or grammar.

20. Discussion: First sentence: Please rephrase it. It needs to be clearer.

21. Page 9: I cannot find "Sub Info 10, Sub Info 11 and 12". Please provide them.

22. It needs to be discussed additionally that different force length scales may alter the outcome of mechanical stimulation. Here, you employed relatively high force levels. Otherwise, it seems to be that all forces applied to animals, cells and tissues are rather bad for the healthy state of animals. The special situation needs to be pointed out here.

Answer to the reviewers

Reviewer #2 (Remarks to the Author):

The involvement of physical stresses in cancer initiation is gaining interest in the research community, but little is known about its importance or mechanisms. The authors use in situ and ex vivo analyses to study the effects of mechanical forces on colon stem cell and tumor dynamics. They conclude that high frequency pulsatile mechanical stresses maintain colon stem cell levels through Ret kinase. They propose that SC levels are pathologically over-stimulated by the physical forces, causing hyperproliferation and contributing to tumorigenesis. The work is comprehensive and potentially significant. However, there are a few issues that need to be addressed to improve the manuscript and support the conclusions.

We thank the reviewer for his/her positive comments on the work, and highly constructive suggestions to improve the work and manuscript.

To help the reading of the revised manuscript, modifications are marked in orange in the "Manuscript -marked" manuscript related file.

1. In general, the text is difficult to follow and could benefit from extensive editing.

The text has been systematically edited to simplify its reading, including by cutting the too long sentences as well as by following the reviews editing advice.

2. The experiments involving injection of the Stb-UML particles need further description. What is the fate of these particles after injection and "focusing" using the external magnet? Do they stay in blood vessels or extravasate into tissue? What are the dynamics of their clearance, and how is this affected by the magnetic forces? In the experiments where the magnet is used to localize the UMLs, and then removed, do they later disperse/disappear? And is this different in the animals where the external magnet is left in place?

- Additional experiments were generated to better describe the fate of Stb-UML. UMLs were previously found to have extravasated in the colonic conjunctive tissue, with co-localization with Vimentin mesenchymal cells 30 min after injection, and to be stable in the colon from at least 1 week to 1 month, in the presence of the external stabilizing ("focusing") magnet¹. Here we further characterize that the presence of UMLs in colonic conjunctive tissues, and co-localization with Vimentin in the mesenchymal cells, is stable at least 3.5 months in the presence of the external stabilizing magnet (new Extended Data Figure 8a,b,c). We observe a similar result in CD31 positive endothelial cells (new Extended Data Figure 8a,d). Interestingly, the presence of the stabilizing magnet was required for the presence of injected UMLs into the conjunctive tissue, as observed by the absence of UMLs after injection without magnet (UML 1 m-2m), alike to without UML injection and magnet (Ctrl), and in contrast to with injection and magnet (UML Magnet 1m-2m), in the conjunctive tissue after 1 and 2 months (new Extended Data Fig. 8a,b).

These new data are now described beginning of p5 of the manuscript (in orange in the marked manuscript file).

- Within the conditions when the stabilizing magnet was present but removed after 30 min, similar quantities of UMLs staid stabilised in the conjunctive tissue until 3.5months as well (new Extended Data Figure 2a,b), with similar co-localisation with Vimentin positive mesenchymal cells (new Extended Data Figure 2a,c) – CD31 having not been tested to avoid too much redundancy with Extended Data Figure 8a,d in a manuscript with already many supplementary data.

We additionally find that the UMLs staying in blood vessels are mostly cleared within 3h, and fully cleared at 24h, in both conditions (new Extended Data Figure 2d). Only 6% of the blood cells showed an internalisation of UMLs (the specific cell type could not be distinguished with the cytometry flux methodology here used, due to the limitation of the amount of biological material by condition) and was removed after 24h as well (new Extended Data Figure 2e), in line with the dynamics of 16h of degradation of the UMLs in blood vessels by the liver².

These new data were added last third of p3 of the manuscript, and in associated Sup info 2 (in orange in the marked manuscript file).

3. Along these lines, there should be controls for the stb-UML particles (without magnetic forces applied) in all of the relevant main figures. It appears that many of these critical control groups are included in the extended figures, but they really need to be in the same figures with the magnet application groups.

The stb-UML particles (without magnetic forces applied) controls are now included in all the relevant figures (also those which were previously reported in Extended data figures): Fig. 1g-h and all Fig. 2 a-d.

4. In the videos, the perturbations caused by breathing appear large, and at least comparable to the peristaltic motion. Do the authors have an estimate of how the magnitudes compare? And what about normal ambulatory motion, which may also impinge on the colon?

Other mechanical strains, such as breathing or motion movements may also mechanically affect the colon and add to 2s spontaneous pulsatile stresses, and thus to stem cells level maintenance by mechanotransductive signalling. However, breathing movements, which amplitude compares to colon pulsatile movements (see Video 1), operate in a completely different spectral regime (allowing for efficient filtering in our analysis), and involve a global translation movement of the colon rather than local mechanical deformations of the colon. In addition, these breathing-induced translational movements involve the proximal part of the colon (closer to the thorax, left half of the colon in Videos1,2) rather than the distal one in which all biochemical analysis were performed (see Methods). Finally Win treated mice did daily walk normally compared to non-treated controls. This thus excludes any

interference of breathing movements, or walking movements, in the mechanical stimulation of SC formation in mice and in the present experiments, respectively.

This is newly added as Sup info 11-b of the manuscript (in orange in the marked manuscript file).

Note also that the amplitudes of all those sources of motion are difficult to compare on a quantitative basis as they can be affected by the anesthesia, with differences that may not be correctly easy to evaluate in our setup. Therefore, only the quantitative comparison of one source of motion (e.g. peristaltic motion) pre- and post-WIN injection is informative.

5. The authors should show that the Ret inhibitor Vandetanib is specific, and not causing off-target effects. Another inhibitor or genetic knockout model would help.

Following reviewer's advice, we checked that at the concentration used here, Vande treatment does not repress the activation of its two other targets EGFR and VEGFR2 in the colon of old APC mice with ACF initiating tumorigenesis (Sup info 5 and new Extended Data Fig. 6f,g,h), confirming the specific effect of Vande on Ret activation inhibition.

We furthermore used Danusertib (Danu) as an inhibitor of Ret in addition to Vande (see note* below). We found the inhibition of the mechanical induction of Ret activation, β -cat phosphorylation ($p\beta$ -cat), β -cat cytosolic enrichment, expression of Lgr5 (target of β -cat) and of the hyperproliferative initiation of tumorigenesis (Anomalous crypt formation, ACF) after 1 month of mechanical stimulation *in vivo*, similarly to Vande results (new Extended Data Fig. 17 and new Extended Data Fig. 18a,b). Interestingly, the efficiency of Danu in repressing Ret mechanical activation was slightly lower than the Vande efficiency. And coherently, $p\beta$ -cat, β -cat cytosolic enrichment and Lgr5 expression were repressed with a similar slightly lower efficiency than with Vande as well (see Fig. 4 and Extended data Fig. 14,15, 16 for comparison). We additionally found mechanical induction of Lgr5 expression by direct mechanical stimulation *ex-vivo* (following ref³ methodology) and repressed both by Vande and Danu (new Extended Data Fig. 18c,d), confirming with another and direct mechanical stimulation methodology, the Ret dependence of Lgr5 expression mechanical induction (see note** below).

These new data confirming the specific role of Ret mechanical induction upstream of all of the biochemical and physio-pathological processes here described, from the β -cat pathway activation to Lgr5 expression and tumorigenesis initiation by hyperproliferative ACF formation, are now added in Sup info 5 and beginning of p8 of the manuscript, respectively (in orange in the marked manuscript file).

*Note that we tested two other inhibitors of Ret on preliminary experiments on a few mice before to find Danusertib as the efficient inhibitor (not shown in the manuscript). We began by testing the efficiency of the Blu667⁴ inhibitor in repressing the mechanically induced phosphorylation of Ret ($pRet$) *in vivo*, after 2 hours of mechanical stimulation on 3 mice, which was not efficient in this configuration. We tested the Regorafenib inhibitor in repressing the mechanically induced phosphorylation of Ret ($pRet$) *in vivo*, after 2 hours of mechanical stimulation on 3 mice, with the vehicle of Vandetanib (Vande) (DMSO/Cremophor), which was not efficient in this configuration. We then tested the Regorafenib inhibitor in repressing the mechanically induced phosphorylation of Ret ($pRet$) *in vivo*, after 2 hours of mechanical stimulation on 3

mice, with an alternative vehicle (PPG/PEG400/Poloxamer188), which showed an unexpected stimulating effect of the vehicle alone independently of mechanical stimulation, and prevented its use to specifically test the involvement of pRet mechanical activation in any downstream effect. We finally tested the Danusertib (Danu) inhibitor with Dextrose by injection (IP), which showed a significant repressing effect of mechanical stimulation of pRet *in vivo* within these conditions.

**Due to the lower efficiency of Danu than Vande in repressing the mechanical induction of pRet *in vivo* and downstream processes, we anticipated an effect less sensitive as well in experiments performed with pulsatile mechanical strains (Fig.2d). However, given the significant, but naturally lower sensitivity of the effects due to the less intense pulsatile stimulation on 5 days compared to the more intense permanent stimulation on 1 month (see Sup Info 7), we thus expected difficulties to get to statistically satisfying effects with reasonable number of life mice sacrificed (in Fig.2d, there are already 45 mice sacrificed for the figure, and the less sensitive effect expected is anticipated to require even more for a new experiment). We thus alternatively decided to further test the specificity of Ret in the mechanical induction of Lgr5 expression through the testing of the repressive effect of both Vande and Danu on Lgr5 mechanical induction by direct deformation of explants *ex-vivo*, following the (Whitehead et al 2009)³ procedure. And indeed, we found Lgr5 expression mechanically induced, and repressed by both Vande and Danu (new Extended Data Fig. 18c,d).

6. The overall hypothesis is not clearly stated. Are the authors proposing that the mechanical forces imposed by peristaltic motion in normal physiology cause tumors, or that they enhance tumor growth after the tumor has already been initiated? Or are they saying that tumor growth induces forces that further enhance tumor growth? These concepts are confounded throughout the text.

We thank the reviewer for alerting us on this point. This is in fact the second proposal: we are proposing that the mechanosensitivity of SC formation is involved in the production of the homeostatic rate of SC in response to the physiological pulsed mechanical stress of the colon. And that this rate is pathologically over-stimulated by the presence of additional and more intense (because permanent) anomalous tumour growth pressure in the neighbouring cells of the tumor. Which indeed further enhances tumor growth. We clarified this point second half of p 2 of the manuscript:

“We thus wonder whether pulsatile mechanical stresses could mechanotransductively stimulate the induction of SC that maintain homeostatic renewal levels²⁷ through proliferation, morphogenesis and biochemical patterning of healthy colonic adult epithelial tissue. We additionally wonder if, in the presence of tumors, compressions exerted by associated hyper-proliferative cells lead to an amplification of the physiological pulsed mechanical cues, thereby pathologically amplifying SC number in the healthy tissues compressed by the tumor and creating a permanent positive feedback loop proliferating signal further fueling tumor growth.”

and

“Indeed, we furthermore find that the effect of high frequency pulsatile stresses on the stimulation of physiological SC rate is pathologically over-amplified by additional permanent anomalous tumor growth pressure, leading to a doubling of SC and proliferative cell (PC) number, as well as to the generation of CSC markers in Apc mice, and is thus at the origin of tumorigenic mechanical induction”

This was also clarified in the conclusion of the dedicated results paragraph middle of p 5

“These results show that **tumoral permanent 1kPa pathological mechanical stresses** significantly induces the increase in the SCs number, with final levels of SC amplified by Apc heterozygous mutation and leads to an hyperproliferation state after 1 month.”

an in the general conclusion last third of p 10:

“The enhancement of mechanical stress due to anomalous permanent tumor growth pressure **adding to, and more intense than, physiological pulsatile mechanical stresses,** amplifies this process and leads to hyperproliferative pathological rate of SC...”

7. There are two findings in the study. One shows that the forces increase stem cells, and the other shows that forces enhance tumor formation. More discussion is needed about how these are related.

We have discussed more the link between SC mechanical induction and tumor formation, which was effectively not discussed enough in the previous version of the manuscript, and is now added middle of p 10 of the manuscript following:

“Indeed, in addition to the requirement of an Apc mutation for promoting the mechanical induction of tumorigenesis, a permanent 1kPa pressure applied for months represents an anomalous mechanical environment of higher intensity compared to the physiological 1kPa pulsed mechanical stresses, integrated over time. As the physiological number of SC was found to be mechanosensitively maintained by endogenous pulsed mechanical stresses, the additional permanent tumoral stress should consequently lead directly to a significant increase in SC number, which we observed. This increase in turn should lead to cell hyperproliferation initiating tumorigenesis in compressed tissues, due to the increase of crypt cells renewal dynamics as a direct consequence of the SC number increase that generate proliferative cell (PC) number increase, both of which we also observed. We propose that this process amplifies tumorigenesis by adding mechanical strains leading to more production of SC in healthy tissues compressed by neighboring tumors in the endogenous process of tumor progression.”

Reviewer #3 (Remarks to the Author):

Ho-Boulidoires et al presented an in vivo study about the effect of mechanical stimulation, such as pulsatile stresses, on the presence and proliferation of CS and CSC cells. Moreover, they provide a mechanism for the mechanotransduction process. In specific, they point out to the Ret kinase mechanosensitivity.

We thank the reviewer for his/her positive comments on the work, and highly constructive suggestions to improve the work and manuscript.

To help the reading of the revised manuscript, modifications are marked in orange in the “Manuscript-marked” manuscript related file.

My major comments are listed below:

1. While the entire investigation is quite interesting and new, the manuscript is hardly readable due to long sentences, errors (within the sentences) and complicated description of the results. This should be improved.

The text has been systematically edited to simplify its reading, including by cutting the too long sentences as well as by following the reviews kind editing advice.

2. The pulsed magnetic force stimulation of colonic tissues that have been magnetized with ultrasound-magnetic liposomes seems to be intriguingly. However, I do not understand how the intravenously injected liposomes enter the targeted cells, which seems to be mesenchymal cells in the colon. How do they get there in a targeted manner? Has it been checked where these liposomes are? Are they all only in these cells and why not in blood cells or endothelial cells, since they get in contact with them much earlier? Please discuss these points and provide data how you checked this.

The location of the UML, which we already knew to localize in the conjunctive tissue of the colon, and to co-localize with Vimentin in mesenchymal cells from 30 min to 1 week at least in the presence of the stabilizing small magnet¹, was here further characterized following the reviewer questions.

- Additional experiments were generated to better describe the fate of Stb-UML. UMLs were previously found to have extravasated in the colonic conjunctive tissue, with co-localization with Vimentin mesenchymal cells 30 min after injection, and to be stable in the colon from least 1 week to 1 month, in the presence of the external stabilizing (“focusing”) magnet¹. Here we further characterize that the presence of UMLs in colonic conjunctive tissues, and co-localization with Vimentin in the mesenchymal cells, is stable at least 3.5 months in the presence of the external stabilizing magnet (new Extended Data Figure 8a,b,c). We observe a similar result in CD31 positive endothelial cells (new Extended Data Figure 8a,d). Interestingly, the presence of the stabilizing magnet was required for the presence of injected UMLs into the conjunctive tissue, as observed by the absence of UMLs after injection without magnet (UML 1 m-2m), alike to without UML injection and magnet (Ctrl), and in contrast to with injection and magnet (UML Magnet 1m-2m), in the conjunctive tissue after 1 and 2 months (new Extended Data Fig. 8a,b).

These new data are now described end of beginning of p5 of the manuscript (in orange in the mark manuscript file).

- Within the conditions when the stabilizing magnet was present but removed after 30 min, similar quantities of UMLs staid stabilised in the conjunctive tissue until 3.5 months as well (new Extended Data Figure 2a,b), with similar co-localisation with Vimentine positive mesenchymal cells (new Extended Data Figure 2a,c) – CD31 having not been tested to avoid too much redundancy with Extended Data Figure 8a,d in a manuscript with already many supplementary data.

We additionally find that the UMLs staying in blood vessels are mostly cleared within 3h, and fully cleared at 24h, in both conditions (new Extended Data Figure 2d). Only 6% of the blood cells showed an internalisation of UMLs (the specific cell type could not be distinguished with the cytometry flux methodology here used, due to the limitation of the amount of biological material by condition) and was removed after 24h as well (new Extended Data Figure 2e), in line with the dynamics of 16h of degradation of the UMLs in blood vessels by the liver².

The process through which the stabilizing magnet favours extravasation through localized magnetic forces applied to UMLs circulating into micro-vessels is not known and will be the object of future investigations.

These new data were added last third of p3 of the manuscript, and in associated Sup info 2 (in orange in the mark manuscript file).

3. Page 2 Line 5 from bottom: "This makes of high frequency..."Please rephrase. What does it mean here? (It needs to be past.)

Thank you. This has been rephrased the following: "Thus, high frequency pulsatile stress cues act as a new niche for SC, through the Ret/ β -cat mechanosensitive pathway".

4. Page 3 Second Para: "submitted" seems not to be the right term here. Please change. Thank you. This has been rephrased the following: "we first applied to Lgr5-GFP mice the cannabinoid receptor"

5. Page 3 Line 11 from bottom: The sentence "Pulsation of ...motor driven... by a motor". Thank you. This has been rephrased the following: "Pulsations of magnetically induced 1kPa pressure were thus ensured by the oscillation of a magnetic checkerboard driven by a motor, between 2.5cm and 22cm every 2s for 5 days (Fig. 1e, Video3)."

6. Page 3 Line 7 from bottom: There are several long sentences, such as "Movements detectable.." which needs to be rephrased. Thank you. This has been rephrased the following: "Movements detectable on the overall colon organ are of 2s period. Their amplitude is consistently smaller of two order of magnitudes than endogenous pulsatile ones, because the 1kPa stress is here magnetically produced directly in the conjunctive epithelial tissue that includes epithelial crypts only. Indeed, the latter is ten times thinner than the visceral smooth muscle from which the 1kPa stress produces the endogenous peristaltic movements imposed to the conjunctive and crypt tissues³⁵ (see Sup Info 3d)."

7. Figure 1e is hardly readable (alter the colors of the text). The colors of the text has been changed in black to make it better readable, thank you.

8. How do I know where the liposomes are in the mice when injected into the blood system?

Are they marked with a fluorescent dye to follow their uptake? Which cell type harbor these liposomes? Please provide an image. The liposomes are marked with a Rhodamine fluorochrome engrafted on PE phospholipids and are thus followed in fluorescence¹. This information was indeed missing, thank you. It is now added at the beginning of the new description of UML fate after injection middle of p3:

“UML are labelled with a Rhodamine fluorochrome engrafted on PE phospholipids and detectable by fluorescence⁸.”

Into the colon, UML are present in the conjunctive tissue of the crypts and co-localize with Vimentin positive mesenchymal and CD31 positive endothelial cells (Extended Data Figure 2a-c and Extended Data Figure 8a,c-d). In the blood, they are at 94% found into the plasma, and at 6% present in indetermined cells (Extended Data Figure 2d,e). See point 2 of the review for more details. These new data were added last third of p3 of the manuscript, and in associated Sup info 2 (in orange in the mark manuscript file).

9. Page 4 Fourth Para. it should be mentioned again what WIN-defective means here. Do you mean WIN treatment? It needs to be consistent throughout the manuscript. Thank you. This has been rephrased the following: “WIN-treated pulsatile-defective”, consistently with the “WIN-treated” expression used in the other parts of the manuscript.

10. Page 4 Para 5: Please correct the sentence "Consistently, we did.... where (?)". Thank you. This has been rephrased the following: “And indeed, no rescue was observed in UML-loaded colon crypts of WIN-treated mice in the absence of pulsed stress (Fig. 2a,b).”, and moved end of Sup Info 4.

11. Page 4 Last Para: "By magnetic cells applying..." needs to be two sentences (hard to read). Thank you. This has been rephrased the following: “We thus magnetically applied the 1kPa pressure with a small strong magnet localized on the skin in front of the colon after UML intravenous injection, that have been previously quantitatively measured⁸ and shown to simulate hyper-proliferative Anomalous Crypt Foci (ACF) formation that initiate tumorigenesis, after one month⁸. Here we found UML present and stable in colonic conjunctive tissues from 1 week⁸ and up to 3,5 months in the presence of the magnet (Extended Data Fig. 8a,b).”

12. Page 5 First Para: Which Notch is meant here? There are several types. Please explain why this specific one is selected then. Please provide the antibody employed (it is not in the list or provided elsewhere). It is also not given in "Extended Data Fig 7a-d".

There exists indeed 4 Notch paralogues in mice, Notch 1,2, 3 and 4, with Notch 1 and Notch2 only being expressed into gut track epithelial cells. Notch 1 and Notch 2 have redundant proliferative functions, as shown by knock out, but Notch1 is far prominent compared to Notch2⁶. In addition, Notch 1 is up-regulated in colon adenocarcinoma⁷. Notch 1 is thus a relevant marker of proliferative cells, and of hyperproliferation in mice colon cancer.

This is now clarified middle of p 5, and “Notch” was replaced by “Notch 1” in the overall manuscript.

Because no Notch1 antibody is available, membrane associated GFP (mG) is used to label Notch1 expressing cells, via the Notch1 promotor driven activation of the “Cre-Lox” system inducing colonic expression of mG conditioned by tamoxifen injection in Notch1-CreERT2/Rosa26mTmG mice ⁶ (see scheme below). GFP expression was here followed by immunofluorescence labelling.

Scheme: Red: Rosa26mTmG strain: pCA is a CMV (cytomegalovirus) enhancer. pA are polyadenylation sequences that stabilize RNAm. They are ended by a Stop codon. Black triangles represent Lox target sites for Cre-recombinase enzyme mediated recombination. Blue”Cre”: Notch1-CreERT2: Cre expression is under the control of Notch1 promotor, but CreERT2 stays into the cytoplasm and does not trigger recombination into the nucleus. After Tamoxifen induction: Tamoxifen induces CreERT2 nuclear translocation and recombination. The mT (membranar-targeted Tomato red fluorescent protein) is excised by Notch1 dependent expression of Cre and allows the pCA enhancer to drive the expression of the membrane-targeted green fluorescent protein (mG) in Notch1 expressing cells. Adapted from ⁸.

This was described at the beginning of Methods section p14 of the previous version of the manuscript (line 3 of Experimental model and subject details), with the GFP antibody listed p18, but was effectively unclear into the text. This is now clarified middle of p5 the following : « As Notch 1 anti-body was not available, we monitored Cre-Lox conditioned expression of membrane GFP induced by the Notch1 promotor⁶ (see Methods).” And in the beginning of p15 section Methods related section: “GFP expression was here followed by immunofluorescence labelling”.

13. Page 5 Para 3: What means "Both of them..." there are listed more than two. Please clarify. Thank you. “Both of them “ was replaced by “All of them”.

14. Page 6: It is not explained why Paneth niche cells are investigated (no connection to the former part). Yes. Paneth cells are part of known gastric SC niche cells, and are tested as such in this paragraph. Unless rarely found in WT mice colon, they are found in tumorous human colon tissues, which is the reason why we tested the possibility of a mechanical stimulation of SC production. This was better detailed end of p6 beginning of p7 following:

“Paneth cells also sustain an essential SC niche in mice intestinal crypts⁵⁵. More rarely found in colon crypts⁵⁴⁻⁵⁶, they are abnormally observed in human colon cancer tissues⁵⁷. We thus checked whether Paneth cells could be present as SC niche cells mechanotransductively induced by tumor growth pressure into the colon.”

15. Page 7: "Sup Info9": I am sorry, but I cannot find it. Is it "Extended Data Fig. 9"? Many apologies for this. It was accidentally cut. Here it is re-introduced as the new Sup info 10- following: " **10-** 1kPa tumor growth pressure stimulates tumor-initiator ACF formation after 1 month¹ sensitively enough to check the effect of Vande on ACF mechanical induction, with a treatment duration no longer than one month. However, the ACF, that are detectable in colonoscopy with a blue coloration, are sporadically induced, and cannot be localized by eye in post-surgery explants, which prevents their targeted observation by histological cuts."

16. Page 7 First Para Line 8: "We found beyond that..."Please rephrase this sentence to increase its clearness. Thank you. The sentence has been rephrased middle of p7 following: "Here we find that the mechanical activation of Ret by tumor growth pressure is upstream of the β -cat target gene Lgr5 in SC and CSC *in vivo*, thanks to the use of the Vande specific inhibitor of Ret mechanical induction of phosphorylation in mice colon (Fig. 4a,c)"

17. Page 8 Second Para: "Consistently, p-beta-cat..." needs to be rephrased for clearness. The sentence has been removed following answer to next point 18.

18. Regarding tables 1, 2 and 3, I would show the images as supplementary data or delete this statement in the manuscript. Following the reviewer suggestion, we have deleted this statement to lighten the manuscript.

19. Page 8 Third Para: "We would thus expect for a...". Please cancel "for" and check all other sentences for errors, such as typos or grammar. This is done, thank you: "We would thus expect a generic activation of Ret phosphorylation in most primary solid tumors types. Indeed, Ret was found activated in primary tumors at invasive stage in the pancreas, ovary, lung, head and neck, uveal, breast, endometrium, with a positive tendency for the uterus cervix, with scores varying between 0.2 to 2.2 depending on the organ (Fig. 7c).", second part of p8.

20. Discussion: First sentence: Please rephrase it. It needs to be clearer. We have clarified the sentence following: "The presence of a tumor generates abnormal permanent mechanical stresses that consist of a mechanotransductive activator of pathological pathways and tumorigenic processes. These can act inside the tumor^{7,60}, as well as outside the tumor in the healthy tissue compressed by tumor growth⁸.", beginning of p9.

21. Page 9: I cannot find "Sub Info 10, Sub Info 11 and 12". Please provide them. Many apologies for this. It was accidentally cut. Here it is re-introduced as new Sup Info 11-a,12,13 following:

"**11-a-** The Ret positive cells are found into the Lgr5 expressing domain of the crypts bottom, as well as slightly upper. Note that the role of such a small number of mechanically induced Ret positive cells and crypts in Lgr5+ cells expression suggests the existence of a transient dynamical mechanical

activation of Ret in a few cells of the 3D crypts, or of few crypts, that would affect all crypts overtime. A hypothesis to be experimentally tested in future work.

12- Note that in a Wnt partially deficient context, an anomalous too low number of SC was found to biochemically favor Apc deficient cells fixation in the crypts and to genetically stimulate tumorigenesis in the intestine¹⁶. We here find that a tumor growth pressure mechanically induces anomalously high level of SC that favors hyperproliferative tumor-initiating ACF formation in the colon. This interestingly indicates the importance of a fine-tuned regulation of SC number to avoid genetically or mechanically induced tumorigenic processes.

13- To directly anticipate for pre-clinic and potential future clinic applications, and as the pharmacological approach is here demonstrated to be specific to Ret mechanical activation (Extended Data Fig.6), pharmacological tools were chosen to synergically address both the pulsatile underlying origin and treatment potentialities of mechanotransductive tumour progression (Fig. 6 ,7), based on chemical inhibition of Ret activation. “

22. It needs to be discussed additionally that different force length scales may alter the outcome of mechanical stimulation. Here, you employed relatively high force levels. Otherwise, it seems to be that all forces applied to animals, cells and tissues are rather bad for the healthy state of animals. The special situation needs to be pointed out here. **Many thanks for the suggestion. Here it is introduced end of p10 following: “The permanent 1kPa pressure applied for months also represents a relatively high anomalous mechanical context as well compared to benign transient strains that can accidentally be applied to colonic tissues (i.e., transient constipation).”**

- 1 Fernandez-Sanchez, M. E. *et al.* Mechanical induction of the tumorigenic beta-catenin pathway by tumour growth pressure. *Nature* **523**, 92-95, doi:10.1038/nature14329 (2015).
- 2 Thebault, C. J. *et al.* In Vivo Evaluation of Magnetic Targeting in Mice Colon Tumors with Ultra-Magnetic Liposomes Monitored by MRI. *Mol Imaging Biol* **21**, 269-278, doi:10.1007/s11307-018-1238-3 (2019).
- 3 Whitehead, J. *et al.* Mechanical factors activate beta-catenin-dependent oncogene expression in APC mouse colon. *HFSP Journal* **2**, 286-294, doi:10.2976/1.2955566 (2008).
- 4 BLU-667 Controls RET-Altered Thyroid Cancers. *Cancer Discov* **9**, OF5, doi:10.1158/2159-8290.CD-NB2019-084 (2019).
- 5 Clevers, H. Lgr5 Stem Cells in Self-Renewal and Cancer. *Blood* **124**, doi:doi.org/10.1182/blood.V124.21.SCI-40.SCI-40 (2014).
- 6 Fre, S. *et al.* Notch lineages and activity in intestinal stem cells determined by a new set of knock-in mice. *PloS one* **6**, e25785, doi:10.1371/journal.pone.0025785 PONE-D-11-12629 [pii] (2011).
- 7 Reedijk, M. *et al.* Activation of Notch signaling in human colon adenocarcinoma. *Int J Oncol* **33**, 1223-1229 (2008).
- 8 Muzumdar, M. D., Tasic, B., Miyamichi, K., Li, L. & Luo, L. A global double-fluorescent Cre reporter mouse. *Genesis* **45**, 593-605, doi:10.1002/dvg.20335 (2007).
- 9 Sato, T. *et al.* Paneth cells constitute the niche for Lgr5 stem cells in intestinal crypts. *Nature* **469**, 415-418, doi:10.1038/nature09637 (2011).

- 10 van Es, J. H. *et al.* Wnt signalling induces maturation of Paneth cells in intestinal crypts. *Nature cell biology* **7**, 381-386, doi:10.1038/ncb1240 (2005).
- 11 Rothenberg, M. E. *et al.* Identification of a cKit(+) colonic crypt base secretory cell that supports Lgr5(+) stem cells in mice. *Gastroenterology* **142**, 1195-1205.e1196, doi:10.1053/j.gastro.2012.02.006 (2012).
- 12 Clevers, H., Loh, K. M. & Nusse, R. Stem cell signaling. An integral program for tissue renewal and regeneration: Wnt signaling and stem cell control. *Science (New York, N.Y.)* **346**, 1248012, doi:10.1126/science.1248012 (2014).
- 13 Weaver, V. M., Howlett, A. R., Langton-Webster, B., Petersen, O. W. & Bissell, M. J. The development of a functionally relevant cell culture model of progressive human breast cancer. *Semin Cancer Biol* **6**, 175-184, doi:10.1006/scbi.1995.0021 (1995).
- 14 Samuel, M. S. *et al.* Actomyosin-mediated cellular tension drives increased tissue stiffness and beta-catenin activation to induce epidermal hyperplasia and tumor growth. *Cancer Cell* **19**, 776-791, doi:S1535-6108(11)00166-8 [pii] 10.1016/j.ccr.2011.05.008 (2011).
- 15 Fernandez-Sanchez, M. E. *et al.* Mechanical induction of the tumorigenic beta-catenin pathway by tumour growth pressure. *Nature* **523**, 92-95, doi:10.1038/nature14329 (2015).
- 16 Huels, D. J. *et al.* Wnt ligands influence tumour initiation by controlling the number of intestinal stem cells. *Nature communications* **9**, 1132, doi:10.1038/s41467-018-03426-2 (2018).

Reviewers' comments:

Reviewer #2 (Remarks to the Author):

The authors have done a good job responding to the previous review. However, I have a few additional concerns and follow-ups:

1. What are the Vimentin-positive cells that the nanoparticles accumulate in? If they are macrophages or fibroblasts, do the particles or exogenous forces affect their biology, inducing fibrosis or inflammation that could also contribute to the observations?
2. There is still some confusion about the overall message. If tumor-generated pressure is needed for tumorigenesis, then where does the tumor-inducing stress come from initially? On the other hand, if tumor-induced stress increases stem cell proliferation, what effect does this have on the already growing tumor? These questions could be addressed by inducing the adenoma as is figure 7, and then removing the magnet in some mice. Does the tumor then grow more slowly than a tumor where additional, exogenous stress is maintained? This could also be tested more systematically on syngeneic tumors implanted in the colon, by applying pressure and measuring tumor growth or animal survival. Again, the authors have shown that the pressure affects SCs, but the relevance of this to tumor initiation and/or tumor growth are not clear.
3. The use of the phrase "tumor growth pressure" is potentially misleading. The authors should replace this with "magnetically-simulated growth pressure" when they are referring to their exogenous perturbation rather than endogenous tumor-generated pressure.

Reviewer #3 (Remarks to the Author):

All my points are addressed very well. The manuscript has improved greatly and the authors put a lot of additional work in it to satisfy the reviewers concerns.

Answer to the reviewers

Reviewer #2 (Remarks to the Author):

The authors have done a good job responding to the previous review.

We do thank the reviewer for his/her positive feedback on the revisions.

However, I have a few additional concerns and follow-ups:

Thank you for the very interesting following-up questions.

1. What are the Vimentin-positive cells that the nanoparticles accumulate in? If they are macrophages or fibroblasts, do the particles or exogenous forces affect their biology, inducing fibrosis or inflammation that could also contribute to the observations?

The presence of UML into Vimentin positive cells of the conjunctive mesenchyme is stable for at least 3.5 months (Extended Data Fig.8a,c). This *a priori* excludes its presence in activated macrophages due to any inflammatory response, in which it should be rapidly eliminated and could not be stabilized for months, and indicates that UML are stabilized into the mesenchymal cells of the conjunctive tissue of the colon, including fibroblasts. In addition, the presence of UML *per se* (*i.e* in the absence of magnet applied for one month) does not show any inflammatory or fibrotic anatomopathological phenotype⁸, excluding any UML-induced inflammation and fibrosis that would participate to the Ret dependent stimulation of SC and hyperproliferation in epithelial cells in addition to mechanical stimulation.

Finally, Ret activation is observed 1 minute only after mechanical stimulation *ex-vivo*, and consistently at the time UML stabilization (30min after injection in the presence of the stabilization magnet) in the mesenchyme *in vivo*⁸. Such one-minute time scale, in line with the mechanotransductive activation of Ret, is too short to activate any inflammatory response that should take at least 4 days⁷⁷, or of fibrotic protein expression, protein production taking at least several hours into the colon⁵⁹. No inflammatory or fibrotic response to the presence of UML *per se*, or to magnetic forces applied by the magnet on the UML stabilized into the mesenchymal cells of the conjunctive tissue of the colon, can thus participate to the initiation of the activation of Ret that leads to the stimulation.

This is newly added as new Sup Info 8, newly referred first paragraph of p5 of the main text.

2. There is still some confusion about the overall message. If tumor-generated pressure is needed for tumorigenesis, then where does the tumor-inducing stress come from initially? Again, the authors have shown that the pressure affects SCs, but the relevance of this to tumor initiation and/or tumor growth are not clear.

Sporadic tumors are classically initiated by mutations, which in Apc heterozygous mice (representative of 85% of colon tumors in humans^{30,31}) is thought to be due to the full loss of APC expression^{70,71} after sporadic loss of heterozygosity (LOH) in the 16-months old mice

here studied. We thus propose that, subsequently and as a second step, the stress developed by the genetically initiated tumor growth stimulates and fuels tumor progression through mechanical induction of additional SC formation in neighboring healthy tissues*.

This is now added at the end of the discussion, beginning of p11. The fact that spontaneous sporadic tumors are initiated by genetic alterations was added everywhere needed into the manuscript, including at the end of the abstract and in the introduction.

*This is indeed supported into the manuscript both by the Ret-dependent mechanical stimulation of SC formation leading to hyperproliferation and of CSC markers expression in initially non-tumorous tissues in which the tumor growth pressure produced by a neighboring tumor was mimicked with magnetic forces (Fig. 3,4 and Extended 9 and 11,12), and by the inhibition of spontaneous genetically initiated hyperproliferative domains, CSC markers and tumors growth by blocking the Ret mechanosensitive pathway involved, in the 16 months-old predisposed Apc mutated mice (Fig. 6c-f). Which is directly relevant to tumor progression.

On the other hand, if tumor-induced stress increases stem cell proliferation, what effect does this have on the already growing tumor? These questions could be addressed by inducing the adenoma as is figure 7, and then removing the magnet in some mice. Does the tumor then grow more slowly than a tumor where additional, exogenous stress is maintained? This could also be tested more systematically on syngeneic tumors implanted in the colon, by applying pressure and measuring tumor growth or animal survival.

Removing the magnet in Fig 6c to follow the adenoma evolution cannot be done in this experiment in which the adenoma is spontaneously produced in the Apc old mice without magnetic force. Moreover, setting up and performing an experiment modulating magnetic forces in implanted tumor would require a couple of years, with an uncertain outcome would tumor growth pressure saturate the mechanotransductive response and make growth insensitive to additional pressure. We thank the reviewer for this suggestion, and evocated this point as a suggestion for future experiments end of Sup Info 14: "These tools" (Ret inhibitors) "can thus be used in future preclinic studies with PDOX implantation, with possible coupling to magnetic pressure stimulation".

3. The use of the phrase "tumor growth pressure" is potentially misleading. The authors should replace this with "magnetically-simulated growth pressure" when they are referring to their exogenous perturbation rather than endogenous tumor-generated pressure.

Thank you. We thus systematically used "magnetically-mimicked growth pressure" when referring to exogenous perturbation rather than endogenous tumor-generated pressure, as systematically underlined into the manuscript.

Reviewer #3 (Remarks to the Author):

All my points are addressed very well. The manuscript has improved greatly and the authors put a lot of additional work in it to satisfy the reviewers concerns

We do thank the reviewer for his/her positive feedback on the revisions.

REVIEWERS' COMMENTS:

Reviewer #2 (Remarks to the Author):

The authors have addressed my remaining issues.